# ALIGNING GENERATIVE DENOISING WITH DISCRIMINATIVE OBJECTIVES UNLEASHES DIFFUSION FOR VISUAL PERCEPTION

**Ziqi Pang**[*]    **Xin Xu**[*]    **Yu-Xiong Wang**
University of Illinois Urbana-Champaign
{ziqip2, xinx8, yxw}@illinois.edu

## ABSTRACT

With success in image generation, generative diffusion models are increasingly adopted for discriminative scenarios because generating pixels is a unified and natural perception interface. Although directly re-purposing their generative denoising process has established promising progress in specialist (*e.g.*, depth estimation) and generalist models, the inherent gaps between a generative process and discriminative objectives are rarely investigated. For instance, generative models can tolerate deviations at intermediate sampling steps as long as the final distribution is reasonable, while discriminative tasks with rigorous ground truth for evaluation are sensitive to such errors. Without mitigating such gaps, diffusion for perception still struggles on tasks represented by multi-modal understanding (*e.g.*, referring image segmentation). Motivated by these challenges, we analyze and improve the alignment between the generative diffusion process and perception objectives centering around the key observation: *how perception quality evolves with the denoising process*. (1) Notably, earlier denoising steps contribute more than later steps, necessitating a tailored **learning objective** for training: *loss functions should reflect varied contributions of timesteps* for each perception task. (2) Perception quality drops unexpectedly at later denoising steps, revealing the sensitiveness of perception to *training-denoising distribution shift*. We introduce *diffusion-tailored data augmentation* to simulate such drift in the **training data**. (3) We suggest a novel perspective to the long-standing question: why should a generative process be useful for discriminative tasks – *interactivity*. The denoising process can be leveraged as a controllable **user interface** adapting to users' correctional prompts and conducting multi-round interaction in an agentic workflow. Collectively, our insights enhance multiple generative diffusion-based perception models *without* architectural changes: state-of-the-art diffusion-based depth estimator, previously underplayed referring image segmentation models, and perception generalists. Our code is available at https://github.com/ziqipang/ADDP.

## 1 INTRODUCTION

The success of diffusion models (Ho et al., 2020; Ramesh et al., 2022; Rombach et al., 2023) has gone beyond pure image generation recently. They emerge as attractive candidates for perception models (Gan et al., 2024; Ke et al., 2024) with the prior knowledge stored in their pre-trained weights and the vision of *generalist* models via unifying various perception tasks under the same pixel generation interface. Recent approaches finetune pre-trained diffusion models and have made promising progress in both state-of-the-art depth estimation (Ke et al., 2024) specialist and generalist models (Gan et al., 2024) supporting tasks from geometric depth estimation to semantic segmentation and detection. However, they commonly focus on designing the generative format without investigating the fundamental gaps between the generative diffusion process and discriminative tasks: the generative process aims at *sampling reasonable distributions*, while discriminative tasks require *precise matches* with rigorous ground truth. Without addressing such discrepancies, generative diffusion models noticeably under-perform when perception tasks involve intricate multi-modal reasoning, *e.g.*, referring image segmentation (RIS), which consequently constrains the exploration of generative perception (Geng et al., 2023). Therefore, the investigation of this paper aspires to bridge the gaps between the generative denoising process and discriminative perception.

---

[*]Equal contribution.

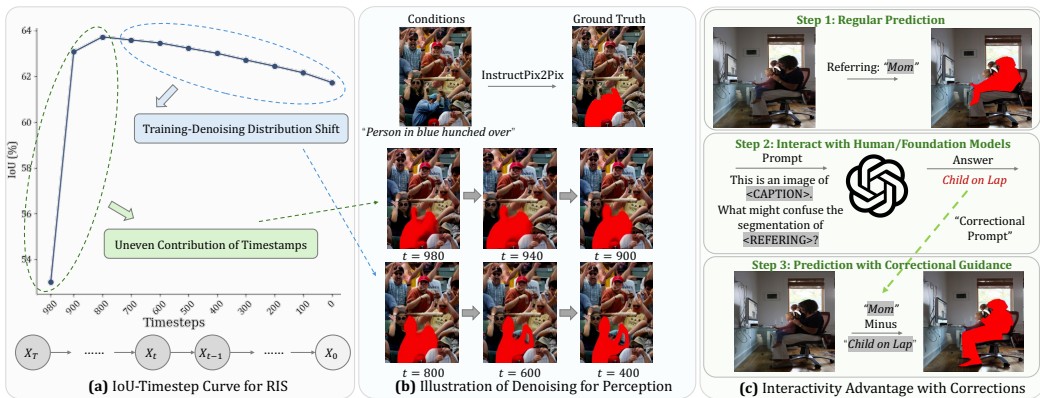

Figure 1: We demonstrate the gaps between a generative denoising process and perception tasks using referring image segmentation (RIS), where the diffusion model learns to color the referred object with red masks. **(a)(b)** The perception quality (Intersection-over-Union, IoU) at intermediate denoising steps, which come from the same denoising trajectory, reveals the *uneven contribution of timesteps* and *training-denoising distribution shift*, addressed by our enhanced **learning objective** and **training data**. **(c)** We discover that the generative denoising process is also a unique **user interface** for discriminative perception, because of its capabilities to *interact with the correctional guidance* from users or foundation models.

In principle, the most profound difference between a generative denoising process and a conventional discriminative model is the *iterative sampling procedure*, where a diffusion model gradually approximates the final prediction by sampling from a score function (Song et al., 2021) step by step. However, such an intuition does not align with the reality of perception. As an intuitive illustration, we choose the challenging RIS task and inspect how the perception quality evolves during the denoising process (Fig. 1). Following previous diffusion-based perception (Gan et al., 2024; Geng et al., 2023), our model adopts an image editing format and specifies RIS as editing the target region, *i.e.*, the objects referred by the language prompt, to red masks. Ideally, the denoising timesteps should gradually refine these red masks to distinguish the target object, but their IoU (Fig. 1a) and appearances (Fig. 1b) unveil the opposite: (1) the *contribution of timesteps is significantly uneven*; and (2) *perception quality drops surprisingly at later denoising steps*. Centering around these observations, we align the *training* of the denoising process with the reality of the *sampling* process in diffusion-based perception models, including the **learning objective** and **training data**.

The *uneven contribution of denoising steps* (Fig. 1a) motivates us to enhance the **learning objective** of diffusion models by reflecting the perception contribution of every timestep in the loss functions. In conventional diffusion training, *e.g.*, DDPM (Ho et al., 2020), the timesteps are treated *uniformly* to learn *single-step* denoising. However, perception tasks need to minimize the distance between the ground truth and *accumulation of multi-step* denoising, which necessitates enhancing the training of more critical steps. Moreover, the surprising *decrease of perception quality* (Fig. 1b) arises from the *training-denoising distribution shift*, which is unique under the diffusion-based perception context: deviated distribution from sampling steps can still produce reasonable images, but they are *wrong* for discriminative tasks with rigorous ground truth. To train diffusion models that are robust to such distribution shift, we leverage *data augmentations* to simulate the erroneous intermediate denoising steps by purposefully corrupting the ground truth. Such improvement to **training data** addresses distribution shifts in the denoising process and maintains the perception quality until later steps.

Finally, we suggest a novel perspective to the long-standing question: *how can the stochastic generative process be useful for discriminative tasks*? This becomes an increasingly important question when diffusion models are used as feature extractors (Zhao et al., 2023) *without* the denoising process. Instead, we propose that the generative process enables an *interactive* and *interpretable* **user interface**. Specifically, a diffusion model can be guided by the *correctional prompts* from users to adjust their predictions progressively (as in Fig. 1c) with classifier-free guidance (Ho & Salimans, 2022), which is a rare ability for conventional single-step discriminative models. For example, our diffusion model enables using language as the multi-round reasoning interface for RIS in an agentic workflow (Ng, 2024) built from GPT4 (Achiam et al., 2023).

To conclude, we have made the following contributions to align the generative denoising process in diffusion models for perception:

1. **Learning objective.** We illustrate the *uneven contribution across denoising timesteps* and reveal such importance by specifying the sampling weights of timesteps accordingly.
2. **Training data.** We demonstrate the *training-denoising distribution shift* and introduce *diffusion-tailored data augmentation* to effectuate especially the later denoising steps.
3. **User interface.** We suggest the unique *interactivity* advantage of diffusion models for perception: they can progressively modify the predictions via the correctional prompts from humans or foundation models, which is essential for an agentic workflow and human-involved applications.

Our insights are collectively named "ADDP" (**A**ligning **D**iffusion **D**enoising with **P**erception). Its enhancements generalize across diverse generative diffusion-based perception models, including state-of-the-art diffusion-based depth estimator Marigold (Ke et al., 2024) and generalist InstructCV (Gan et al., 2024). Our ADDP also extends the usability of diffusion-based perception to multi-modal referring image segmentation, where we enable a diffusion model to catch up with *some* discriminative baselines for the first time. We hope ADDP overcomes the limitations of generative diffusion-based perception and unlocks new opportunities in this domain.

## 2 PRELIMINARIES

**Diffusion Models.** Diffusion models have been analyzed in multiple formulations (Ho et al., 2020; Karras et al., 2022; Song et al., 2021), and here we adopt the DDPM (Ho et al., 2020) style since most diffusion-based perception models are implemented in DDPM's way. In DDPM, diffusion models learn the image distribution $P(x)$ via a reverse Markov chain with length $T$. It gradually denoises a random variable $x_T$, which commonly follows Gaussian distribution, into the target variable $x_0$. During training, the model learns a denoising objective $\varepsilon$ with a neural network $\varepsilon_\theta(\cdot)$,

$$t \sim \text{Uniform}(\{1,...,T\}), \ \varepsilon \sim \mathcal{N}(0, \mathbf{I}), \ x_t = \sqrt{\bar{\alpha}_t} x_0 + \sqrt{1-\bar{\alpha}_t}\varepsilon, \ \mathcal{L} = \mathbb{E}_{(x_t, t, \varepsilon)} ||\varepsilon - \varepsilon_\theta(x_t, t)||_2^2. \quad (1)$$

To synthesize high-resolution images, recent latent diffusion models, *e.g.*, Stable Diffusion (Rombach et al., 2023), encode images into a latent space for denoising. By training at scale (Schuhmann et al., 2021), these models can integrate text conditions as $\varepsilon_\theta(x_t, t, D)$, where $D$ denotes a language description. Such prior knowledge is the basis of using latent diffusion models for perception.

**Visual Perception Tasks.** We investigate diverse perception tasks and diffusion models to understand the gaps between generative models and discriminative objectives. **(1) Depth Estimation.** We focus on the state-of-the-art diffusion-based Marigold (Ke et al., 2024). Its diffusion model operates as $\varepsilon_\theta(x_t, I, t)$, where $x_t$ is the image latent for depth maps and $I$ is latent of the input image. **(2) Referring Image Segmentation (RIS).** RIS involves an input image $I$ and a referring description $D$ for the target object. We treat RIS as an image editing task and adopt the common framework of InstructPix2Pix (Brooks et al., 2023). Concretely, the objective is to "edit the pixels of the target object to red" via the diffusion model of $\varepsilon_\theta(x_t, I, D, t)$, where $x_t$ is the latent of the image with red segments on the target object. Experimental analysis of our editing formats is in Sec. B.1. **(3) Generalist Perception.** We follow InstructCV (Gan et al., 2024) to build a generalist multi-task perception model for depth estimation, semantic segmentation, and object detection. This generalist model similarly uses InstructPix2Pix to unify diverse tasks into image editing. Concretely, the model operates as $\varepsilon_\theta(x_t, I, D, t)$, where $D$ is the description prompt of the task *e.g.*, "Detect %Category%," and the output $x_t$ is image latent for depth maps, segmentation masks, or bounding boxes.

## 3 METHOD

Motivated by the gaps between the diffusion process and perception objectives (Fig. 1), we propose the corresponding alignments in Fig. 2, which are simple and plug-and-play for diffusion models. (1) Sec. 3.1: improving the **learning objective** by resembling the uneven contribution of timesteps.

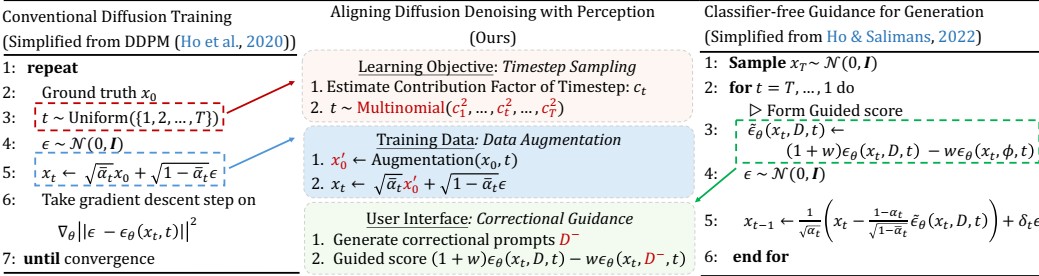

Figure 2: Method overview. We align the generative diffusion models with perception tasks from *learning objective*, *training data*, and *user interface*. Notations follow DDPM (Ho et al., 2020).

(2) Sec. 3.2: simulating the distribution shift by improving the **training data** with data augmentation. (3) Sec. 3.3: re-purposing classifier-free guidance to enable interactive **user interfaces** via the generative denoising process.

## 3.1 LEARNING OBJECTIVE: CONTRIBUTION-AWARE TIMESTEP SAMPLING

We observe the uneven contribution of denoising timesteps in the evolution of perception quality (Fig. 1), *e.g.*, earlier steps closer to $t = T$ have a more significant influence than later steps closer to $t = 0$. This aligns with image generation observations (Zhang et al., 2024b) where earlier steps conduct more influential "semantic planning." However, such properties are not reflected in the learning objective of conventional diffusion training (Eqn. 1): the timesteps $t$ are uniformly sampled, and the optimization targets are all $\varepsilon \sim \mathcal{N}(0, \mathbf{I})$ with similar scales. So why and how should diffusion models tailor their training for distinct timesteps under perception scenarios?

**Necessity of Distinguishing Timesteps in Diffusion Training.** This design is rooted in the different objectives of generative and discriminative tasks. Diffusion models can learn *single-step* score functions for generative tasks to sample *reasonable* images, but discriminative tasks require the final predictions, which are *multi-step integral* of score functions, to *precisely match* rigorous ground truths. To better explain its impact on the learning objective, we define *contribution factors* $c_t$, denoting the *relative contribution* of a timestep $t$ for the final result $x_0$, *i.e.*, $x_0 \propto \sum_{t=1}^{T} c_t \varepsilon_\theta(x_t, t)$[1]. Then the distance between prediction $x_0$ and ground truth $\tilde{x}_0$ is decomposed as below, where $\varepsilon_t$ is the ground truth noise at timestep $t$:

$$\mathbb{E}_{(\tilde{x}_0, x_0)} ||\tilde{x}_0 - x_0||_2^2 \propto \mathbb{E}_{(\tilde{x}_0, x_0, \varepsilon_1, ..., \varepsilon_T)} \sum_{t=1}^{T} c_t^2 ||\varepsilon_t - \varepsilon_\theta(x_t, t)||_2^2. \tag{2}$$

Before discussing the implication of Eqn. 2, we clarify the truncation terms for the right-hand side. (1) $\mathbb{E}_{(\tilde{x}_0, x_0, \varepsilon_1, ..., \varepsilon_T)} \sum_{t_1 \neq t_2} ||(\varepsilon_{t_1} - \varepsilon_\theta(x_{t_1}, t_1))(\varepsilon_{t_2} - \varepsilon_\theta(x_{t_2}, t_2))||_2^2$. As $\varepsilon_t$ is randomly sampled from $\mathcal{N}(0, \mathbf{I})$, the terms $(\varepsilon_t - \varepsilon_\theta(x_t, t))$ are independent, making the whole truncation term zero; thus, this term can be ignored. (2) The intermediate $x_t$ in the denoising process may drift from the precise trajectory $\tilde{x}_t$ for precisely sampling the ground truth $\tilde{x}_0$. Our approximation ignores the truncation errors caused by this drift since it is intractable over iterative sampling. With these analyses, we proceed with the right-hand side of Eqn. 2 to improve the diffusion loss functions.

A natural implication from Eqn. 2 is that the contribution $c_t^2$ needs to be reflected in the training loss of each timestep $\mathcal{L}_t = ||\varepsilon_t - \varepsilon_\theta(x_t, t)||_2^2$. This does not contradict the original diffusion objective since the model still learns to fit the score function on single timesteps. However, the errors from more influential timesteps are penalized more in our way, aligning better with the perception objective. As a side note, this principle is consistent with how rectified flow models re-weight the loss functions of timesteps to guide the optimization (Kingma & Gao, 2024). However, we additionally offer guidelines to *utilize* and *estimate* the values of $c_t^2$ for specific perception objectives. Concretely, one can: (1) *scale the loss values* by $c_t^2$, or (2) *scale the sampling probability* of timesteps with a multinomial distribution using $c_t^2$ as sampling weights for $t$ (Fig. 2). Both variants improve diffusion for perception, but the *probability scaling* method performs better (Sec. 4.3.1). The following parts discuss how to estimate the values of $c_t^2$.

**Deriving $c_t^2$ from Diffusion Formulation.** To start with, $c_t^2$ is an inherent property of diffusion models since each step $\varepsilon_\theta(x_t, t)$ can be converted to $x_0$ following DDPM (Ho et al., 2020):

$$x_0 = \frac{1}{\sqrt{\bar{\alpha}_t}} x_t - \frac{\sqrt{1 - \bar{\alpha}_t}}{\sqrt{\bar{\alpha}_t}} \varepsilon_\theta(x_t, t). \tag{3}$$

Therefore, we can interpret $\frac{\sqrt{1 - \bar{\alpha}_t}}{\sqrt{\bar{\alpha}_t}}$ as the relative importance of timestep $t$, indicating how much of $x_0$ can be explained by $\varepsilon_\theta(x_t, t)$. Then we normalize them to acquire $c_t^2 = (\frac{1 - \bar{\alpha}_t}{\bar{\alpha}_t}) / \sum_{i=1}^{T} (\frac{1 - \bar{\alpha}_i}{\bar{\alpha}_i})$.

**Deriving $c_t^2$ from Perception Statistics.** The above derivation is discussed in terms of the latents $x_t$ (Eqn. 2). However, we find the contribution of timesteps sensitive to the perception tasks and diffusion models. For example, Marigold (Ke et al., 2024) for depth estimation exhibits a smoother precision ($\delta_1$) curve during denoising (Fig. 3) compared with the IoU in RIS (Fig. 1a). Therefore, $c_t^2$ becomes a unique property for each diffusion-based perception scenario and motivates us to estimate each task statistically with the same principle of $c_t^2$: *how much of the final prediction can*

---

[1]We use $\propto$ here to accommodate varying noise schedulers and the normalization of estimating $c_t$.

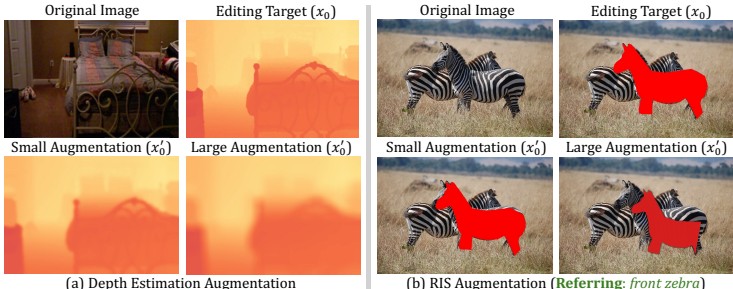

Figure 4: Data augmentation of **(a)** Gaussian blurring for depth estimation, and **(b)** color / shape / location for RIS. We use large / small intensities of augmentations to simulate different scales of distribution shifts at the earlier / later steps of denoising.

*be explained by an intermediate denoising step*? Concretely, we can derive this via the *coefficient of determination* in regression analysis, denoted as $R^2$, which measures the goodness of a fit. This procedure involves three steps. (1) *Data collection*. We apply a diffusion-based perception baseline to $N$ validation samples and get $N \times T$ metric values of intermediate denoising steps. Without loss of generality, we take IoU from RIS as an example and acquire $\{\text{IoU}_{t,i}\}_{t \leq T, i \leq N}$. (2) *Initial regression*. We initialize the estimation with the first step $c_T^2$ by running a linear regression of $\text{IoU}_{0,:} = \beta + \beta_T \text{IoU}_{T,:}$. The $R^2$ value of this regression, denoted as $(R^2)_T$, is the proportion of the final IoU explained by the first denoising step, so we adopt it as $c_T^2$. (3) *Iterative estimation*. We iteratively add new timesteps into the regression model and set $c_t^2 \leftarrow (R^2)_t - (R^2)_{t+1}$, indicating the increase in explained IoU with the new timestamp. Please note that $(R^2)_t - (R^2)_{t+1}$ is

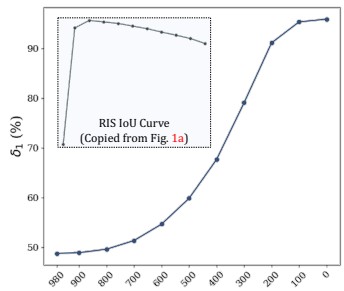

Figure 3: Evolution of $\delta_1$ (intuitively the "accuracy" for depth estimation) from Marigold (Ke et al., 2024) shows smoother patterns than RIS. We copy the RIS curve from Fig. 1a here for easier comparison.

non-negative since adding new variables can only improve the regression fit. More discussions and implementation details are in Sec. A.2.

## 3.2    TRAINING DATA: DIFFUSION-TAILORED DATA AUGMENTATION

In RIS scenarios, we observe the unexpected IoU drop at later denoising steps (Fig. 1a), where the masks gradually deviate from correct regions and show hallucinated shapes (Fig. 1b). This reveals the *training-denoising distribution shift*: $x_t$ during training (Eqn. 1) comes from the ground truth, while it might deviate from the ideal sampling path during inference. Generative model studies (Ning et al., 2023; 2024) call this "exposure bias." However, it is even more critical for discriminative scenarios: shifted $x_t$ might still produce reasonable images belonging to the *distribution* of ground truth but no longer fit the desired ground truth of that sample precisely.

**Necessity of Simulating Distribution Shift.**    The ideal solution for distribution shift is to train the diffusion models with $x_t$ sampled from the actual denoising process. However, this is computationally infeasible due to the iterative nature of denoising. Therefore, we take a step back and introduce the solution of *simulating the distribution shift for training* with augmentations to the ground truth.

**Diffusion-tailored Data Augmentation.**    We purposefully corrupt the ground truth $x_0$ into $x_0'$ so that the $x_t$ for training (Eqn. 1) reflects distribution shift. Such corruption depends on the timesteps by using more intense augmentation for earlier timesteps: Intuitively, the perception results are coarser at the initial denoising and should be simulated with larger deviations from the real ground truth. When incorporated into the training pipeline of DDPM, the procedure becomes:

$$x_0' = \text{Augment}(x_0, t), \quad \varepsilon \sim \mathcal{N}(0, \mathbf{I}), \quad x_t = \sqrt{\bar{\alpha}_t} x_0' + \sqrt{1 - \bar{\alpha}_t} \varepsilon. \tag{4}$$

We design different augmentations to capture the typical distribution shift for each task as in Fig. 4. For instance, the RIS format is a red mask, so our designed augmentation involves color (color changes), location ( transformations to masks), and shape (random erasing of mask parts); while depth estimation mimics coarse boundaries and adopts Gaussian blur. As critical implementation details, we discover that $x_0$-prediction of diffusion models are more suitable for such data augmentation than its mathematically equivalent $\varepsilon$-prediction, and the benefits of varying augmentation intensities *w.r.t* timesteps. More details are discussed in Sec. A.3.

**Discussion: Distinctions with Conventional Data Augmentation.**    Our data augmentation significantly improves the performance and effectuated later denoising steps (Sec. 4.3). Moreover, our

insights are different from conventional data augmentation. Compared with perception studies, *e.g.*, RIS, cannot adopt augmentation since masks close to image borders disable "random cropping," and referring with "up/down/left/right" disable "random flipping." Compared with diffusion studies, we extend the boundary of the common practice of merely training on ground truth denoising trajectories – the diffusers $\varepsilon_\theta(\cdot)$ can explicitly learn to correct problematic input into precise prediction.

### 3.3 User Interface: Interactivity via Correctional Guidance

We suggest a novel perspective on the value of a diffusion denoising process for discriminative tasks: with perception tasks intrinsically deterministic, *why and how would the generative sampling in diffusion models be helpful*? This long-standing problem is increasingly important with emerging studies using pre-trained diffusion models as *single-step* generative models or feature extractors (Parmar et al., 2024; Xu et al., 2024), without leveraging the *multi-step* generative process. Besides the vision of unifying perception into pixel synthesis, we demonstrate that a generative model can serve as an interactive user interface for perception, which is especially critical for human-involved applications and beyond the capabilities of conventional discriminative models.

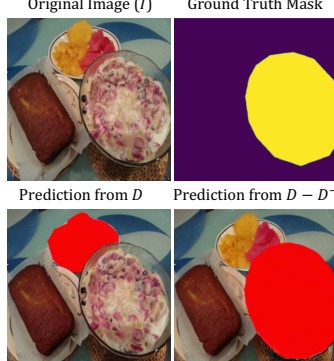

Original Image (*I*)          Ground Truth Mask

Prediction from *D*          Prediction from $D - D^-$

**Referring** (*D*):
*A glass bowl of food sitting on a whicker holder*
**Correctional Prompt** ($D^-$):
*White plate with yellow and red fruits*

Figure 5: Interacting with correctional guidance $D^-$.

**Interactivity via Denoising.** Diffusion models can be expressed with a score-matching formulation (Song et al., 2020), where $\varepsilon_\theta(x_t, t)$ matches a score function $\nabla_{x_t} \log p(x_t)$. This provides a natural way to control the generation with *explicit user guidance* from the interaction with humans or foundation models. Multi-modal understanding, *i.e.*, the previously overlooked RIS in diffusion-based perception, is an ideal application to demonstrate the unique interactivity value. We show an intuitive example in Fig. 1c.

**Correctional Guidance via Classifier-free Guidance.** Our approach is based on classifier-free guidance (Ho & Salimans, 2022): given a condition $C$, adding $(\varepsilon_\theta(x_t, t, C) - \varepsilon_\theta(x_t, t, \phi))$ ($\phi$ denotes empty condition) to the original prediction can *nudge* the diffusion result to align better with $C$. In the case of RIS, we consider both conditions of referring description $D$ and image $I$. If the model makes an error and the user can specify the error with language descriptions $D^-$, the original prediction can be corrected by adding $(\varepsilon_\theta(x_t, t, D, I) - \varepsilon_\theta(x_t, t, D^-, I))$ (Fig. 5). Here, we refer to $D^-$ as *correctional guidance*, similar to negative prompts (Podell et al., 2024) but grounded in perception and vision-language reasoning. Based on the compositionally of score functions (Liu et al., 2022), we utilize the correctional guidance $D^-$ in addition to the image guidance term of $w_I$:

$$\tilde{\varepsilon}_\theta(x_t, t, D, D^-, I) = \varepsilon_\theta(x_t, t, \phi_D, \phi_I) + w_I \Big( \varepsilon_\theta(x_t, t, \phi_D, I) - \varepsilon_\theta(x_t, t, \phi_D, \phi_I) \Big)$$
$$+ w_D^- \Big( \varepsilon_\theta(x_t, t, D^-, I) - \varepsilon_\theta(x_t, t, \phi_D, I) \Big) \tag{5}$$
$$+ w_D \Big( \varepsilon_\theta(x_t, t, D, I) - \varepsilon_\theta(x_t, t, D^-, I) \Big).$$

$w_D^-$ and $w_D$ are scalers for the correctional guidance strength. By setting $w_D > w_D^-$, Eqn. 5 increases the margins between denoising from $D$ and $D^-$. More discussion and details are in Sec. A.4.

**Integration with Agentic Workflows.** To validate the value of such a user interface at a large scale, we construct an agentic workflow (Ng, 2024) with GPT4o (Achiam et al., 2023) to automatically generate the correctional prompts $D^-$. Concretely, we propose a *two-round* proof-of-concept workflow. (1) Use LLaVA (Liu et al., 2023c) to provide a detailed caption of the original image $I$. (2) A language-only GPT4o guesses top-$k$ confusing objects in the image based on the referring $D$ and image caption. (3) Our diffusion model generates $k$ new predictions from each correctional prompt using Eqn. 5, and applies a majority vote to produce a final mask. More details are in Sec. A.4.

## 4 Experiments

### 4.1 Datasets and Implementation Details

Our insights are generalizable for diffusion-based perception and cover diverse scenarios. **(1)** We improve the state-of-the-art diffusion-based Marigold (Ke et al., 2024) for depth estimation, following the same zero-shot evaluation setups. (2) We investigate RIS, where previous diffusion-based models show large gaps to discriminative counterparts. We follow the standard practice by fine-tuning an

| Method | ETH3D AbsRel↓ | ETH3D $\delta_1$↑ | ScanNet AbsRel↓ | ScanNet $\delta_1$↑ | NYUv2 AbsRel↓ | NYUv2 $\delta_1$↑ | Diode AbsRel↓ | Diode $\delta_1$↑ | KITTI AbsRel↓ | KITTI $\delta_1$↑ | Average Rank |
|---|---|---|---|---|---|---|---|---|---|---|---|
| DiverseDepth (Yin et al., 2020) | 22.8 | 69.4 | 10.9 | 88.2 | 11.7 | 87.5 | 37.6 | 63.1 | 19.0 | 70.4 | 7.4 |
| MiDaS (Ranftl et al., 2020) | 18.4 | 75.2 | 12.1 | 84.6 | 11.1 | 88.5 | 33.2 | 71.5 | 23.6 | 63.0 | 7.1 |
| LeReS (Yin et al., 2021) | 17.1 | 77.7 | 9.1 | 91.7 | 9.0 | 91.6 | 27.1 | 76.6 | 14.9 | 78.4 | 5.1 |
| Omnidata (Eftekhar et al., 2021) | 16.6 | 77.8 | 7.5 | 93.6 | 7.4 | 94.5 | 33.9 | 74.2 | 14.9 | 83.5 | 4.7 |
| HDN (Zhang et al., 2022) | 12.1 | 83.3 | 8.0 | 93.9 | 6.9 | 94.8 | 24.6 | **78.0** | 11.5 | 86.7 | 3.1 |
| DPT (Ranftl et al., 2021) | 7.8 | 94.6 | 8.2 | 93.4 | 9.8 | 90.3 | **18.2** | 75.8 | **10.0** | 90.1 | 3.8 |
| Marigold (Ke et al., 2024) | 7.1 | 95.1 | 6.9 | 94.5 | 6.0 | 95.9 | 31.0 | 77.2 | 10.5 | 90.4 | 2.4 |
| +ADDP (Ours) | **6.3** | **96.1** | **6.3** | **95.6** | **5.6** | **96.3** | 29.6 | 77.5 | **10.0** | **90.6** | **1.4** |

Table 1: Comparison with diffusion-based depth estimator Marigold (Ke et al., 2024) with identical pre-training and zero-shot generalization to real-world benchmarks. **Bold** numbers are the best, underscored are the second best. Our method ADDP uses *contribution-aware timestep sampling* ("Sampling") and *diffusion-tailored data augmentation* ("Aug") and consistently improves Marigold across these scenarios.

| Method | RefCOCO val | RefCOCO test-A | RefCOCO test-B | RefCOCO+ val | RefCOCO+ test-A | RefCOCO+ test-B | G-Ref val | G-Ref test |
|---|---|---|---|---|---|---|---|---|
| *Discriminative Encoder-Decoder Based* | | | | | | | | |
| MCN (Luo et al., 2020) | 62.44 | 64.20 | 59.71 | 50.62 | 54.99 | 44.69 | 49.22 | 49.40 |
| EFN (Feng et al., 2021) | 62.76 | 65.69 | 59.67 | 51.50 | 55.24 | 43.01 | 51.93 | - |
| VLT (Ding et al., 2022) | 65.65 | 68.29 | 62.73 | 55.50 | 59.20 | 49.36 | 52.99 | 56.65 |
| ReSTR (Kim et al., 2022b) | 67.22 | 69.30 | 64.45 | 55.78 | 60.44 | 48.27 | - | - |
| CRIS (Wang et al., 2022) | 70.47 | 73.18 | 66.10 | 62.27 | 68.08 | 53.68 | 59.87 | 60.36 |
| LAVT (Yang et al., 2022) | 72.73 | 75.82 | 68.79 | 62.14 | 68.38 | 55.10 | 61.24 | 62.09 |
| VPD (Zhao et al., 2023) | 73.25 | - | - | 62.69 | - | - | 61.96 | - |
| ReLA (Liu et al., 2023a) | 73.82 | 76.48 | 70.18 | 66.04 | 71.02 | 57.65 | 65.00 | 65.97 |
| PVD (Cheng et al., 2024) | 74.82 | 77.11 | 69.52 | 63.38 | 68.60 | 56.92 | 63.13 | 63.62 |
| UNINEXT (Yan et al., 2023) | 77.90 | 79.68 | 75.77 | 66.20 | 71.22 | 59.01 | 70.04 | 70.52 |
| *Generative Image Synthesis Based* | | | | | | | | |
| Unified-IO (Lu et al., 2022) | 46.42 | 46.06 | 48.05 | 40.50 | 42.17 | 40.15 | 48.74 | 49.13 |
| InstructDiffusion (Geng et al., 2023) | 61.74 | 65.20 | 60.17 | 46.57 | 52.32 | 39.04 | 51.17 | 52.02 |
| InstructPix2Pix-SD1.5 | 60.87 | 63.70 | 58.39 | 44.98 | 51.93 | 35.31 | 43.99 | 45.43 |
| + ADDP (Ours) | 66.86 | 67.39 | 63.72 | 55.35 | 58.72 | 48.45 | 55.85 | 57.05 |
| InstructPix2Pix-SD2.0 | 64.96 | 66.72 | 62.63 | 47.13 | 53.32 | 38.99 | 50.28 | 50.58 |
| + ADDP (Ours) | **69.14** | **70.27** | **67.46** | **57.58** | **61.65** | **51.67** | **59.05** | **59.60** |

Table 2: RIS Comparison. Our insights collectively mitigate the gaps between generative and discriminative ones by large progress. Although not achieving the state of the art, our improvements empower the common diffusion baseline, *i.e.*, RIS finetuned InstructPix2Pix, to catch up with *some representative* discriminative baselines *for the first time*. We hope this improved baseline removes the constraints and encourages new opportunities for perception with generative diffusion models.

InstructPix2Pix (Brooks et al., 2023) model on RefCOCO (Yu et al., 2016), RefCOCO+ (Yu et al., 2016), and G-Ref (Nagaraja et al., 2016) separately for 60 epochs. (3) We follow InstructCV (Gan et al., 2024) and prove the effectiveness of our insights under a multi-task generalist setting, where a single model addresses depth estimation, semantic segmentation, and object detection. Due to space limits, the detailed training and evaluation setups for these experiments are in Sec. A.

## 4.2 MAIN RESULTS

### 4.2.1 DEPTH ESTIMATION

Diffusion-based perception methods are already effective for depth estimation, represented by the recent Marigold (Ke et al., 2024). Although only trained on synthetic depth, Marigold performs competitively in a zero-shot way. We apply both of our improvements on learning objectives (Sec. 3.1) and training data (Sec. 3.2) to Marigold and show the quantitative comparison following Marigold's style in Table 1. Notably, our proposed techniques consistently improve Marigold across *all the benchmarks*. We conduct detailed ablations of the two techniques for depth estimation in Sec. B.4.

### 4.2.2 REFERRING IMAGE SEGMENTATION (RIS)

**Improvement of Generative RIS.** We format RIS as an image editing problem and separately train on RefCOCO, RefCOCO+, and G-Ref, for a fair comparison with previous studies. As in Table 2, we focus on the comparisons with *generative methods* and include discriminative approaches for context. We first emphasize the *significant challenge of RIS for generative perception methods*, despite their strong performance in other tasks like depth estimation. This limitation constrains the development of generative perception research. Note, our claim is not on achieving state-of-the-art RIS performance. Rather, we demonstrate that our ADDP, with its plug-and-play insights, substan-

Figure 6: Interactive interface enables diffusion models to adaptively correct their predictions via language models' guidance $D^-$. Such capabilities of *progressiveness* are beyond conventional discriminative models and are an emerging advantage of the generative denoising process in perception.

| Method | NYUv2 (Depth Estimation) RMSE↓ | ADE20K (Semantic Segmentation) mIoU↑ | COCO (Object Detection) mAP@0.5↑ |
|---|---|---|---|
| InstructCV | 0.302 | 46.67 | 46.6 |
| +ADDP (Ours) | **0.288** | **48.40** | **48.1** |

Table 4: Generalist Perception. We follow InstructCV (Gan et al., 2024) and build a multi-task generalist perception model using InstructPix2Pix without task-specific components. Our techniques show consistent improvement across these three fundamental perception tasks.

tially narrows the gap between generative and discriminative methods, enabling a common diffusion framework InstructPix2Pix to catch up with *some* RIS baselines *for the first time*, without modifying the model or introducing extra data. We hope our enhanced diffusion-based method inspires further exploration of generative perception in tackling multi-modal understanding challenges.

From Table 2, we have the following conclusions. **(1)** Compared with other *generative* methods, especially the baseline of InstructPix2Pix, we significantly and consistently *improve all the RIS* subsets through the integration of better learning objective (contribution-aware timestamp sampling) and training data (diffusion-tailored data augmentation). **(2)** Compared with the few existing methods adapting pre-trained image generative models for perception, we outperform them by a large margin without tailoring the model architecture or using more data for RIS, especially another InstructPix2Pix-based method – InstructDiffusion (Geng et al., 2023). These indicate that our discovered insights are critical for designing generative perception models and open new opportunities for diffusion-based perception. **(3)** Our insights generalize across different stable diffusion models (Rombach et al., 2023), enhancing them by a large margin. More ablations are in Sec. 4.3.

**Effectiveness of Generative Denoising as Interactive User Interfaces.** To validate the benefits of interactivity (Sec. 3.3) and support the value of the generative process for perception, we evaluate our proof-of-concept agentic workflow with correctional guidance on the validation sets of RIS,

Please note that our Table 2 does not use such interactive reasoning to avoid unfair comparison with other RIS methods. As in Table 3, our workflow can improve the grounding and *gain larger advantages on harder scenarios* (G-Ref), especially the most challenging G-Ref. In Fig. 6, we further illustrate qualitative examples of how our generated correctional prompts modify the grounding results

| Method | RefCOCO | RefCOCO+ | G-Ref |
|---|---|---|---|
| InstrucPix2Pix | 60.87 | 47.14 | 50.28 |
| +Sampling+Aug (Ours) | 66.86 | 55.35 | 55.85 |
| +Correctional Guidance (Ours) | **66.93** | **56.13** | **56.98** |

Table 3: Effectiveness of correctional guidance, especially on hard scenarios (G-Ref).

via the reasoning conducted by language models. These results indicate that the interactive interface of diffusion models is beneficial for perception tasks involving reasoning or user interaction.

### 4.2.3 GENERALIST PERCEPTION

A key motivation for using diffusion models for perception is the vision of building generalist perception models by unifying diverse tasks into image generation. We follow InstructCV (Gan et al., 2024) in this endeavor and solve three fundamental perception tasks simultaneously: depth estimation, semantic segmentation, and object detection. They are formatted as image editing, addressed with the InstructPix2Pix (Brooks et al., 2023) framework. As shown in Table 4, our ADDP consistently improves the InstructCV using vanilla InstructPixPix across all three tasks. This validates the effectiveness of our approach for broader diffusion-based perception models.

### 4.3 ABLATION STUDIES

We analyze the effectiveness of our insights through a series of ablation studies on the most challenging RIS task. Without special mention, the experiments are conducted on the RefCOCO benchmark with 20 epochs of training as Sec. 4.1 and evaluation on the RefCOCO's validation set. More ablation studies are provided in Sec. B.

### 4.3.1 CONTRIBUTION-AWARE TIMESTAMP SAMPLING

In Table 5, we analyze the strategies proposed in Sec. 3.1: enlarging the contribution of earlier denoising steps in learning objectives. Specifically, we compare four strategies: (1) *Uniform*: the original DDPM strategy, where the timesteps are uniformly sampled, and the losses are not scaled; (2) *Loss Scaling (Diffusion)*: scaling the loss of a timestep by $c_t^2$ estimated from the diffusion formulation. (3) *Prob Scaling (Diffusion)*: Sampling the timesteps by $t \sim$ Multinomial$(c_1^2, ..., c_T^2)$, where $c_t^2$ is derived from the diffusion formulation. (4) *Prob Scaling (Perception Stats)*: Sampling the timesteps with the $c_t^2$ estimated from the perception (IoU) statistics. As shown in Table 5, reflecting the contribution of timesteps in either sampling or loss weights enhances the uniform baseline. With $c_t^2$ from diffusion weights, scal-

| Method | $c_t^2$ | oIoU |
|---|---|---|
| Uniform | N/A | 56.42 |
| Loss Scaling | Diffusion | 58.21 |
| Prob Scaling | Diffusion | 63.05 |
| Prob Scaling | Perception Stats | **64.00** |

Table 5: Strategies of using the contribution $c_t^2$ in diffusion training.

ing the sampling probability is better than scaling the loss, which is likely due to that $\varepsilon_\theta(x_t, t)$ is trained with more iterations at earlier denoising steps under the probability scaling. Moreover, $c_t^2$ estimated from the perception tasks performs the best, since this is most closely aligned with the objective. These comparisons support our design in Sec. 3.1 to improve the learning objective of diffusion for perception.

### 4.3.2 DIFFUSION-TAILORED DATA AUGMENTATION

**Effectiveness of Data Augmentation.** In Fig. 7, we compare the IoU-Timestep curves before and after applying data augmentation. Specifically, we calculate the IoU at the 2nd, 20th, 40th, 60th, 80th, and 100th sampling out of 100 denoising steps in to-
tal. Fig. 7 validates our diffusion-tailored data augmentation from two aspects. **(1)** The quality of masks significantly improves. Such an increase mostly comes from the earlier denoising steps, indicating the benefits of providing more challenging inputs to the diffusion model and enforcing the model to correct the errors. **(2)** The trend of IoU-Timestep curve shows that IoU keeps increasing slowly after $t = 800$, contrasting the decrease of InstructPix2Pix and "InstructPix2Pix+Sampling" between $t = 800$ and $t = 200$. Despite a subsequent slight drop in metrics at the final stage, our data augmentation largely decreases the overall drops after the early steps. Therefore, our enhanced training data indeed aligns the denoising process with perception tasks by mitigating the training-denoising distribution shifts.

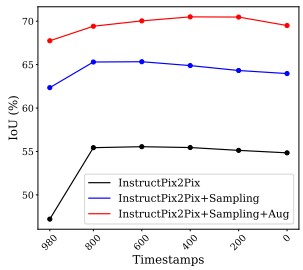

Figure 7: IoU-Timestep curves. Our data augmentation decreases the training-denoising distribution shifts.

**Data Augmentation Intensity.** In Fig. 8, we investigate the relationship between the intensity of data augmentation and final performance. As in Sec. 3.2 and Sec. A.3, the intensity of data augmentation specifies the corruption to the ground truth. For RIS, larger intensities indicate larger changes in ground truth masks' color, location, and shape. We regard the intensity used in Table 2 and Table 5 as the base level ($1\times$), which introduces visually reasonable corruptions. We also evaluate performance under conditions of no augmentation ($0\times$), reduced intensity ($0.5\times$), and increased intensity ($2\times$). As demonstrated in Fig. 8, higher augmentation intensity leads to improved performance, indicating that more intense data augmentation enhances the discriminative capabilities of diffusion models. These findings val-

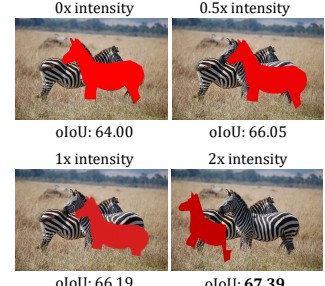

Figure 8: Augmentation Intensity.

idate the effectiveness of our data augmentation strategy. We utilize the median intensity data augmentation in the main experiments, because it visually aligns with our observed data drift during the denoising phase. Further investigation of more intense augmentations will be our future work.

## 5 RELATED WORK

**Diffusion Models.** Diffusion models (Ho et al., 2020; Karras et al., 2022; 2024; Sohl-Dickstein et al., 2015) are probabilistic models denoising from Gaussian noises, guided by a reverse Markovian process. They exhibit better training stability than generative adversarial networks (Esser et al., 2021; Goodfellow et al., 2014) or variational auto-encoders (Kingma & Welling, 2014; Van Den Oord et al., 2017). The recent advances in diffusion models have achieved outstanding text-to-

image synthesis ability (Ramesh et al., 2022), especially with the latent diffusion models represented by the Stable Diffusion series (Esser et al., 2024; Podell et al., 2024; Rombach et al., 2023). The capabilities of generating realistic images with conditioning have motivated numerous applications represented by image editing (Brooks et al., 2023; Hertz et al., 2022; Sheynin et al., 2024) and controllable image generation (Zhang et al., 2023). Besides training stability, diffusion models have the intuition of score-matching functions (Song et al., 2021) and support guidance at the denoising time (Dhariwal & Nichol, 2021; Ho & Salimans, 2022), which is crucial for improving the consistency with conditioning. The success of diffusion models is also progressing quickly in other modalities, such as 3D (Poole et al., 2022) and video generation (Ho et al., 2022). Our study mainly improves such diffusion models under a perception perspective (Gan et al., 2024; Geng et al., 2023; Ke et al., 2024; Xing et al., 2023). Concretely, our insights are plug-and-play alignment to train better diffusion-based perception models, effectuating the generative denoising process for perception objectives. Furthermore, we illustrate how classifier-free guidance (Ho & Salimans, 2022) can be uniquely re-purposed for vision-language reasoning and imply the unique value of generative models for discriminative tasks.

**Diffusion Models for Perception.** Recent studies adopting pre-trained diffusion models for perception, *e.g.*, Stable Diffusion (Rombach et al., 2023), can be categorized into three groups. (1) Diffusion models can synthesize virtual training examples (Nguyen et al., 2024; Tian et al., 2023; 2024; Wu et al., 2023) for perception models. (2) The most profound trend is to leverage pre-trained backbones in diffusion models as feature extractors in perception tasks, supporting tasks like segmentation (Xu et al., 2023; Zhao et al., 2023), depth estimation (Xu et al., 2024; Zhao et al., 2023), 3D understanding (Man et al., 2024), and finding correspondence (Hedlin et al., 2024; Luo et al., 2024; Namekata et al., 2024; Tang et al., 2023; Zhang et al., 2024a). However, these methods do not fully leverage the generative capabilities of diffusion models. (3) Our focus is the last category, which unleashes the generation ability of diffusion models and envisions pixel synthesis as the pivot to developing *generalist models* (Gan et al., 2024; Geng et al., 2023; Xing et al., 2023). Learning perception tasks also improve the precision of generation, *e.g.*, EMU-Edit (Sheynin et al., 2024). Despite the success in depth estimation (Ke et al., 2024), we notice that generative perception remains challenging and inferior to discriminative methods on domains like multi-modal reasoning. Our studies enhance the training and inference of diffusion models by aligning the denoising process with discriminative tasks from the perspectives of learning objectives and training data. Such improvements bring consistent improvement under several scenarios and critically empower competitive diffusion-based baselines for multi-model understanding. In addition, we suggest the unique value of the generative process for visual perception as interactive user interfaces. We hope our discoveries open new opportunities and enable more studies in perception using diffusion models.

## 6 CONCLUSION

This study investigates the missing parts of diffusion models for perception tasks from the fundamental distinction between generative and discriminative tasks: generation requires sampling diverse and reasonable contents, while discriminative perception needs a precise match with the rigorous ground truth. We unveil the gap between the conventional diffusion denoising process and perception tasks and propose plug-and-play enhancements in **learning objective** (contribution-aware timestamp sampling) and **training data** (diffusion-tailored data augmentation). In addition, we highlight the unique advantage of diffusion models as interactive and interpretable **user interface** for perception tasks, empowering multi-round reasoning via agentic workflows. We hope our insights will foster further exploration and improvement of generative models for perception.

**Discussion and Limitations.** We investigate a wide range of diffusion-based perception models and unlock significantly improved baselines. However, we acknowledge that generative perception is inherently challenging: the methods using the denoising process, instead of treating diffusion models as feature extractors, might still underperform on challenging tasks, such as the RIS in our paper. Moreover, the diffusion models are pre-trained for image generation purposes without alignment with perception use cases. For example, Stable Diffusion (Podell et al., 2024; Esser et al., 2024) series employ data filtering to only train on highly aesthetic images, which potentially hurts the generalization to perception data. Therefore, how to guide diffusion models for perception tasks during the *pre-training* stage will be meaningful for future work. Finally, our ADDP methodology could be relevant for generative tasks with relatively well-defined ground truth, such as super-resolution and 3D reconstruction. We hope ADDP can inspire further exploration in such directions.

ACKNOWLEDGMENTS

This work was supported in part by NSF Grant 2106825, NIFA Award 2020-67021-32799, the Toyota Research Institute, the IBM-Illinois Discovery Accelerator Institute, the Amazon-Illinois Center on AI for Interactive Conversational Experiences, Snap Inc., and the Jump ARCHES endowment through the Health Care Engineering Systems Center at Illinois and the OSF Foundation. This work used computational resources, including the NCSA Delta and DeltaAI supercomputers through allocations CIS230012 and CIS240387 from the Advanced Cyberinfrastructure Coordination Ecosystem: Services & Support (ACCESS) program, as well as the TACC Frontera supercomputer, Amazon Web Services (AWS), and OpenAI API through the National Artificial Intelligence Research Resource (NAIRR) Pilot.

ETHICS STATEMENT

The studies conducted in this paper do not have explicit ethics concerns. However, our method potentially shares the social biases of pre-trained diffusion models during data filtering, annotation, and training stages. Therefore, we aim to understand such generative models for perception scenarios and encourage cautious applications with human involvement.

REPRODUCIBILITY STATEMENT

We ensure the reproducibility of all the results in the paper. The implementation details are enumerated in Sec. 4.1 and Sec. A. We have released the code at https://github.com/ziqipang/ADDP.

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

## A  IMPLEMENTATION DETAILS

### A.1  DATASETS AND IMPLEMENTATION DETAILS

#### A.1.1  DEPTH ESTIMATION

We strictly follow the setting in Marigold (Ke et al., 2024) for both training and evaluation. Concretely, the model is trained on the virtual depth maps in Hypersim (Roberts et al., 2021) and Virtual-KITTI (Cabon et al., 2020) with initialization from stable diffusion 2 (Rombach et al., 2023). The evaluation is conducted in a zero-shot style on multiple real-world datasets, including NYUv2 (Silberman et al., 2012), ScanNet (Dai et al., 2017), DIODE (Vasiljevic et al., 2019), KITTI (Geiger et al., 2012), and ETH3D (Schops et al., 2017). The evaluation follows affine-invariant depth evaluation (Ranftl et al., 2020). We adopt the two metrics used in Marigold: Absolute Mean Relative Error (AbsRel) and $\delta_1$ accuracy, which measure the overall errors and precision of depth estimation. For both training and evaluation, we keep Marigold's codebase and hyper-parameters identical. Most importantly, our diffusion model is trained with 1000 steps of DDPM (Ho et al., 2020) and denoised with 50 steps of DDIM (Song et al., 2020). Our improvement in learning objectives and training data are described later in Sec. A.2 (contribution-aware timestep sampling) and Sec. A.3 (diffusion-tailored data augmentation).

#### A.1.2  REFERRING IMAGE SEGMENTATION

**Datasets and Evaluation.**    We follow the standard practice of separately training models on RefCOCO (Yu et al., 2016), RefCOCO+ (Yu et al., 2016), and G-Ref (Nagaraja et al., 2016) (UMD split), which are created from the MSCOCO dataset (Lin et al., 2014), then evaluate on their validation and test sets. We adopt the standard metric of "overall intersection over union" (oIoU). This metric accumulates the intersection and unions across the whole dataset and emphasizes larger objects. Before evaluating, we resize our prediction and ground truth to the resolution of $512 \times 512$ following previous works (Wang et al., 2022).

**Training and Inference.**    Our baseline is InstructPix2Pix (Brooks et al., 2023), taking the referring prompts as the text conditioning and input image as the image conditioning. Our diffusion model is trained with 1000 steps of DDPM (Ho et al., 2020) and denoised with 100 steps of DDIM (Song et al., 2020). During training, we will resize all the images to the resolution of $256 \times 256$, optimizing with the AdamW optimizer (Kingma, 2014; Loshchilov, 2017), batch size of 128, learning rate of $10^{-4}$, and cosine annealing scheduler (Loshchilov & Hutter, 2016). The classifier-free guidance (Ho & Salimans, 2022) weights are manually tuned on the RefCOCO validation set to guarantee optimal performance, which is 1.5 for image conditioning and 7.5 for text conditioning in the InstructPix2Pix model and 1.5 for image conditioning and 3.0 for text conditioning in our enhanced model. The models presented in Table 2 are trained with 60 epochs, where each epoch indicates enumerating each image once. The detailed configurations for our proposed insights (Sec. 3) are described in Sec. A.2 (contribution-aware timestep sampling), Sec. A.3 (diffusion-tailored data augmentation), and Sec. A.4 (correctional guidance).

**Evaluation Post-processing.**    During the evaluation of diffusion for perception, we convert the edited images with red masks into binary masks for IoU computation. Since the generated images might not have a "perfect" red color with RGB values of $(255, 0, 0)$, we apply a color threshold $\delta_c$ to convert the image to a mask.

$$M_{i,j} = \begin{cases} 0, & ||I_{i,j} - (255,0,0)||_2 > \delta_c \\ 1, & ||I_{i,j} - (255,0,0)||_2 \leq \delta_c \end{cases} \tag{A}$$

$I_{i,j}$ and $M_{i,j}$ denote the RGB value of the image and mask value at pixel $(I, j)$. This equation intuitively means that we recognize a pixel as a masked region if its color is close enough (within $\delta_c$) to the perfect red color. In our experiments on the validation set of RefCOCO (Yu et al., 2016), we find that $\delta_c = 50$ is consistently a reasonable threshold. Finally, we clarify that this protocol is only for the convenience of evaluation and is not our major concern.

#### A.1.3  GENERALIST INSTRUCTCV

For the generalist model, we primarily follow the experimental setup of InstructCV (Gan et al., 2024), which adopts InstructPix2Pix (Brooks et al., 2023) to conduct perception in the form of image editing. Although the original InstructCV also conducts image classification, its format of "turning the whole image red if the image matches a certain class" is less intuitive, and we find that including image classification causes large variances in the remaining three tasks. Therefore, we focus on depth estimation, semantic segmentation, and object detection. Specifically, we use NYUv2 (Silberman et al., 2012) for depth estimation, ADE20K (Zhou et al., 2017; 2019) for se-

mantic segmentation, and COCO (Lin et al., 2014) for object detection. We mainly focus on the *contribution-aware timestep sampling* for Table 4, with details in Sec. A.2. We empirically re-weight the ratio of training samples from each dataset to 0.3, 0.3, and 0.4, respectively. The model is trained for 20 epochs, with 100k iterations per epoch.

## A.2 LEARNING OBJECTIVES: CONTRIBUTION-AWARE TIMESTEP SAMPLING

We mainly describe how we estimate and use the $c_t^2$ defined in Sec. 3.1 to improve the training of diffusion models. The list of tasks are in Sec. 2.

### A.2.1 ESTIMATING $c_t^2$ VALUES

**Protocols.** To avoid variance of timesteps, we merge the 1000 steps from DDPM (Ho et al., 2020) into 10 groups, where the 100 timesteps in a group share the same $c_t^2$. **(1)** For the depth estimation using Marigold (Ke et al., 2024), we infer a pre-trained Marigold model on NYUv2 (Silberman et al., 2012) (654 samples) to get the results at intermediate denoising steps. The metric of RMSE is used. **(2)** For RIS task, we use $N = 1,000$ validation samples from RefCOCO (Yu et al., 2016) to estimate the results with *IoU*. The weights from SD1.5 are directly adopted in SD2.0 and SDXL. **(3)** For generalist perception with InstructCV (Gan et al., 2024), we use $N = 500$ validation samples each from NYUv2 (Silberman et al., 2012), ADE20K (Zhou et al., 2017; 2019), and COCO (Lin et al., 2014). We estimate $c_t^2$ for each task separately and adopt the average $c_t^2$ value as the shared sampling weight.

**Estimated $c_t^2$ Values.** In the RIS experiments, we discover that $t = 0$ has an extremely low probability, while the earlier ones closer to $t = T$ have large weights. However, we empirically round up their probabilities to 1% to avoid any timesteps from insufficient training. A comparison of the probabilities between diffusion-based formulation and empirical estimation from perception statistics (RIS and depth estimation) is in Fig. A. As clearly illustrated, **(1)** the weights estimated from IoU statistics are more *unevenly* distributed than diffusion weights, indicating the importance of the first few steps for multi-modal understanding. **(2)** The weights estimated for Marigold and generalist perception are smoother than RIS, which validates that the evolution of perception quality is a unique property for each diffusion-based perception model.

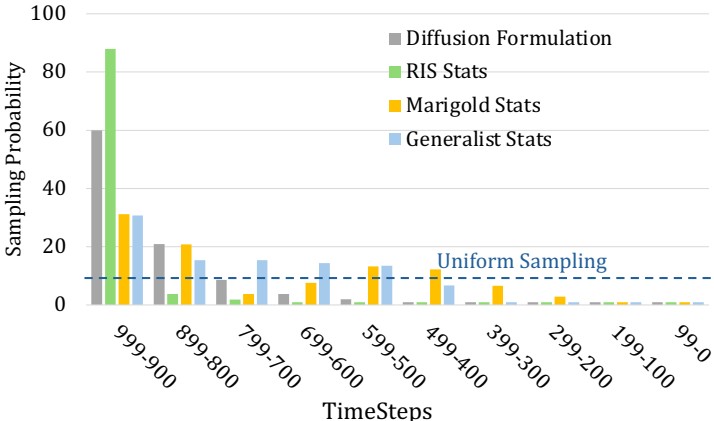

Figure A: Comparison of the sampling weights derived from normalized $c_t^2$, estimated by diffusion formulation or RIS/Marigold/Generalist perception statistics (Sec. 3.1). The dashed blue line denotes the probability of uniform sampling from one of the ten timestamp groups.

### A.2.2 ALGORITHMS

Algorithm A demonstrates the process for estimating the contribution factors $c_t^2$, and Algorithm B illustrates how the contribution factors are utilized during diffusion training via timestamp sampling.

## A.3 DIFFUSION-TAILORED DATA AUGMENTATION

We introduce the detailed implementation of diffusion-tailored data augmentation (Sec. 3.2), which aims at simulating the training-denoising distribution shift during the training time.

---

**Algorithm A** $c_t^2$ Estimation

---

**Require:** DDPM-trained Diffusion-based Perception Model, Perception Quality Metric $Q(\cdot)$.
1: Evaluate on $N$ samples with $T$ intermediate steps, and obtain $\mathscr{D} = \{Q_{t,i}\}_{t \leq T, i \leq N}$
2: $\mathscr{D} \leftarrow \mathscr{D} \cup \{Q_{T+1,i} = 0\}_{i \leq N}$ $\qquad\qquad\qquad\qquad\qquad$ ▷ Dummy initializing
3: Perform linear regression on $Q_{0,:} = \beta + \beta_{T+1} Q_{T+1,:}$, and compute $(R^2)_{T+1}$
4: **for** $t = T, T-1, \ldots, 1$ **do**
5: $\quad$ Perform linear regression on $Q_{0,:} = \beta + \sum_{s=t}^{T} \beta_s Q_{s,:}$ $\qquad$ ▷ $Q_{0,:}$: the final result
6: $\quad$ Compute $(R^2)_t$ $\qquad\qquad\qquad\qquad$ ▷ how well $Q_{i,0}$ is explained by $\{Q_{i,s}\}_{t \leq s \leq T}$
7: $\quad$ $c_t^2 \leftarrow (R^2)_{t+1} - (R^2)_t$
8: **end for**
9: **return** $(c_t^2)_{t=1}^{T}$

---

**Algorithm B** Contribution-aware Timestep Sampling

---

**Require:** Contribution factors $(c_t^2)_{t=1}^{T}$
1: **repeat**
2: $\quad$ $x_0 \sim q(x_0), \varepsilon \sim \mathcal{N}(0, \mathbf{I})$
3: $\quad$ $t \sim \text{Multinomial}(c_1^2, \ldots, c_t^2, \ldots, c_T^2)$
4: $\quad$ Take gradient descent step on $\nabla_\theta \| \varepsilon - \varepsilon_\theta(\sqrt{\bar{\alpha}_t} x_0 + \sqrt{1 - \bar{\alpha}_t} \varepsilon, t) \|^2$
5: **until** converge or reach the maximum iteration

---

### A.3.1 DESIGNS AND IMPLEMENTATIONS

**Depth Estimation.** In the experiments of using Marigold (Ke et al., 2024) for depth estimation, we use Gaussian blur as data augmentation to simulate the rough prediction results at the early denoising steps. Concretely, our Gaussian blur has a kernel size of 31 pixels with the maximum intensity of $10 \times t/T$, where $t$ is the current timestep, and $T = 1000$ is the maximum timestep in DDPM. An intuitive illustration is Fig. 4a.

**RIS.** Considering the potential drift of predictions that can occur during the denoising steps, we apply augmentation on three different aspects during training: *color*, *location*, and *shape*. For color changes, we randomly adjust the color mask's brightness, contrast, and saturation with the maximum intensity of $0.2 \times t/T$. For location changes, we randomly rotate, translate, or scale the mask region. The maximum rotation is 10 degrees, the maximum translation is 0.05 of the image scale, and the scale changes is confined between 0.95 and 1.05 times. For erasing, we randomly crop out parts of the mask with a scale between 0.01 and 0.05. The intuitive demonstration is in Fig. 4b.

We acknowledge that our design of augmentation might not be the optimal one. However, we mainly aim to simulate the imperfect intermediate denoising results and illustrate the broad space of data augmentation for diffusion-based perception.

### A.3.2 PREDICTION TARGET

During the training of diffusion models, it is common practice to adopt Gaussian noise $\varepsilon$ as the objective. However, the integration of data augmentation introduces additional complexity, thereby altering the gradient direction of $\nabla_x p(x)$. To account for the impact of data augmentation, we redefine the training objective as

$$\varepsilon' = \frac{x_t - \sqrt{\bar{\alpha}_t} x_0}{\sqrt{1 - \bar{\alpha}_t}} = \varepsilon + \frac{\sqrt{\bar{\alpha}_t}}{\sqrt{1 - \bar{\alpha}_t}} \left( \text{Augment}(x_0, t) - x_0 \right). \tag{B}$$

Despite this adjustment, empirical results indicate that predicting $\varepsilon'$ is still less effective than predicting $x_0$. A potential explanation for this observation is that correcting the learning target introduces additional discrepancies between the training and denoising phases. In contrast, predicting $x_0$ helps mitigate this issue by directly predicting the final outputs, thereby enhancing model consistency across different phases. The ablation study is presented in Sec. B.3. In addition, we incorporate classifier-free guidance (Ho & Salimans, 2022) when generating $x_0$. In accordance with the formulation in InstructPix2Pix (Brooks et al., 2023), we set $w_D = 3.0$ and $w_I = 1.5$.

### A.4 INTERACTIVITY WITH CORRECTIONAL PROMPTS

This section provides the detailed steps of how we convert the generative denoising process into interactive user interfaces. The details of our evaluation on RIS with an agentic workflow is also discussed.

#### A.4.1 FORMULATION OF CORRECTIONAL PROMPTS

We justify the formulation of our correctional prompts (Sec. 3.3) following the derivation in InstructPix2Pix (Brooks et al., 2023). We treat the correctional prompt $D^-$ as an auxiliary condition alongside the image $I$ and referring $D$, so the objective conditional probability for denoising is:

$$P(x|D,D^-,I) = \frac{P(x,D,D^-,I)}{P(D,D^-,I)} = \frac{P(D,D^-,I|x)P(D^-,I|x)P(I|x)P(x)}{P(D,D^-,I)} \quad (C)$$

By taking the logarithm and derivative of the conditional probability above, we have the score function (Hyvärinen & Dayan, 2005) as below, corresponding to our correctional guidance in Eqn. 5.

$$\nabla_x \log P(x|D,D^-,I) = \nabla_x \log P(x) + \nabla_x \log P(I|x)$$
$$\nabla_x \log P(D^-,I|x) + \nabla_x \log P(D,D^-,I|x). \quad (D)$$

#### A.4.2 PROMPTS AND WORKFLOW

When building the agentic workflow in Sec. 3.3, we leverage the following three steps as depicted in Fig. B, where our correctional guidance votes the masks for right predictions. Please note that the motivation of our workflow is to simulate the rough thinking process of how a human interacts and correct the predictions of referring segmentation.

**(1) LLaVA Captioning.** We first prompt LLaVA (Liu et al., 2023b;c) to create detailed caption of each image, specifically the `llava − v1.6 − vicuna − 13b`, with the prompt showing in Fig. B. The special consideration of our prompt is to pay attention to both foreground and background objects, which both frequently appear in the referring segmentation dataset. After this step, we can input the image descriptions for foundation models to reason.

**(2) Correctional Guidance Generation with GPT.** Then we prompt GPT4 (Achiam et al., 2023) to analyze the captions and referring expressions to name the confusing objects for referring segmentation, which act as the correctional prompts. We acknowledge that this is not an ideal agentic workflow but is computationally feasible within our budget since each validation set of RefCOCO (Yu et al., 2016) has around 10,000 referring phrases. Specifically, we use `gpt − 4o − 2024 − 05 − 13` to conduct the prompts in the middle of Fig. 5. We explicitly use examples to guide GPT into reasoning the confounding objects for referring segmentation while providing the arguments. The output of the GPT will contain three confounding objects, namely the correctional prompts.

**(3) Referring Mask Prediction.** We independently predict the referring masks for each of the $k=3$ correctional prompts, applying guidance weights of $w_I = 1.5$, $w_D^- = 2.0$, and $w_D = 3.0$ in accordance with Eqn. 5. The final mask is derived from pixel-wise majority voting across the $k=3$ masks, as depicted in Fig. B.

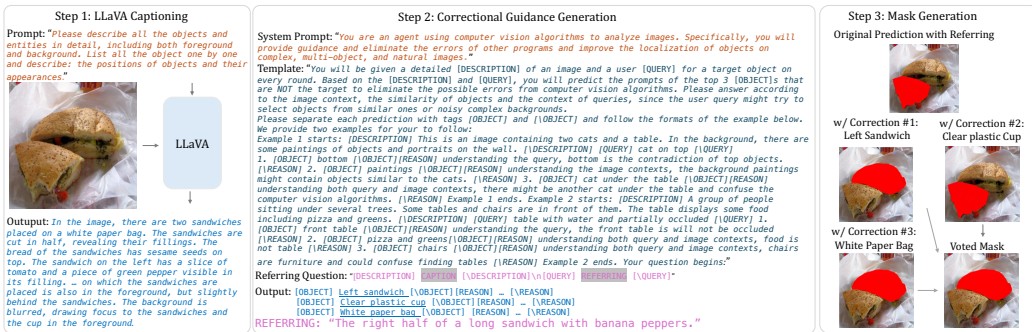

Figure B: Our workflow of generating correctional prompts shows the advantage of the interactivity of diffusion-based perception.

# B  ADDITIONAL ABLATION STUDIES

## B.1  REFERRING IMAGE SEGMENTATION MASK FORMAT

As described in Sec. 2, we format referring image segmentation (RIS) as an image editing problem: painting the regions of target objects with solid red masks. This section analyzes why we chose red masks in our experiments.

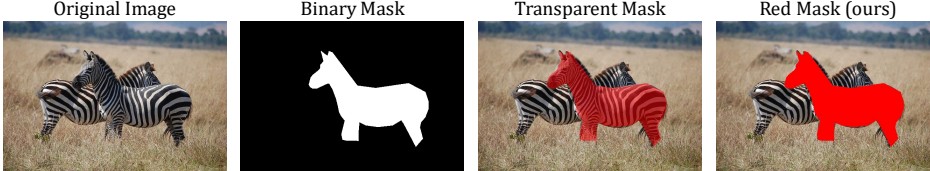

Figure C: Comparison of mask encoding methods for RIS. Note that ground-truth masks are used here for demonstration purposes.

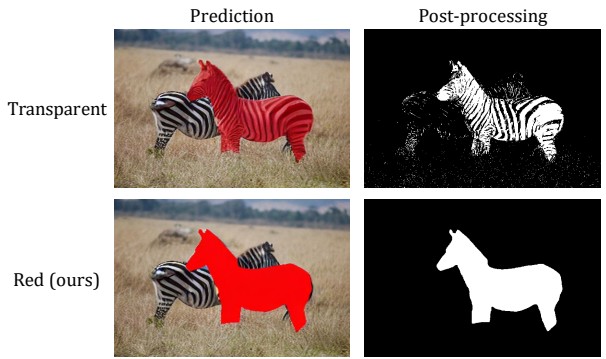

Figure D: Comparison of post-processing. The solid red mask simplifies the post-processing for evaluation, which also encourages us to adopt it as the RIS format.

**Solid or Transparent Masks.**   Previous studies that have adopted different formats: InstructDiffusion (Geng et al., 2023) utilizes transparent masks, InstructCV (Gan et al., 2024) employs binary masks with RGB channels, while our strategy is a solid red mask, as visually compared in Fig. C. When trained with InstructPix2Pix (Brooks et al., 2023) for 20 epochs, the use of binary masks results in a sub-optimal result, achieving an oIoU of 28.18%, significantly lower than the 56.42% obtained by red masks. This performance gap is likely attributable to the lack of contextual information in the generated binary masks, which makes the grounding harder. As presented in Table 2, our red masks (InstructPix2Pix) achieve performance comparable to transparent masks (Instruct-Diffusion) with fewer training iterations, thereby underscoring the effectiveness of red masks. In addition, red masks facilitate simpler post-processing via straightforward pixel-wise thresholding, whereas applying thresholding to transparent masks produces excessively noisy results, as shown in Fig. D. Although incorporating an additional U-Net (Ronneberger et al., 2015) for segmentation post-processing might enhance the results (Geng et al., 2023), this introduces additional complexity to the evaluation process. Therefore, given the effectiveness and efficiency, we choose to use red masks in our experiments.

**Color of Masks.**   We additionally analyze the influence of colors on the RIS performance. These experiments train the baseline InstructPix2Pix and our enhancements with 40 epochs on RefCOCO. As shown in Table A, our insights consistently improve RIS for all the scenarios. The red masks perform the best, so we chose it as our default setting. We hypothesize that the influence of color is similar to the discovery in visual prompt tuning (Shtedritski et al., 2023): distribution of images affect model's performance under different colors of visual prompts.

|  | Red | Blue | Green |
|---|---|---|---|
| InstructPix2Pix | 60.15 | 53.88 | 53.54 |
| +Sampling+Aug | **66.51** | **65.87** | **65.72** |

Table A: Comparison of mask colors for RIS.

## B.2 Intermediate Denoising Results

As discussed in Sec. 4.3, with contribution-aware timestep sampling and diffusion-tailored data augmentation, we observe a less pronounced decrease in IoU. Since the final outputs of diffusion models are typically obtained at timestep $t=0$, we base our main results on timestep $t=0$ for fair comparisons in Table 2. However, as indicated by Fig. 7, the IoU at timestep $t=200$ is the highest among all timesteps. Hence, we also report results from this timestep in Table B, which show slightly better predictions than those at timestep $t=0$. Further bridging this gap indicates better alignment between the denoising process and perception objective, and will be the future work.

| Timestamp | RefCOCO | | | RefCOCO+ | | | G-Ref | |
|---|---|---|---|---|---|---|---|---|
| | val | test-A | test-B | val | test-A | test-B | val | test |
| $t=0$ | 66.86 | 67.39 | 63.72 | 55.35 | 58.72 | 48.45 | 55.85 | 57.05 |
| $t=200$ | **67.34** | **68.36** | **64.44** | **56.12** | **59.87** | **49.06** | **57.41** | **57.90** |

Table B: Comparison on the oIoU of RIS at intermediate denoising steps. The SD1.5-based RIS models from Table 2 are used for this comparison.

## B.3 Training Data: Data Augmentation for Diffusion Models

$x_0$**-prediction v.s. $\varepsilon$-prediction.** As mentioned in Sec. A.3.2, using corrupted input as data augmentation (Sec. 3.2) is sensitive to the prediction target of the scheduler in diffusion models. For example, the default $\varepsilon$-prediction in DDPM (Ho et al., 2020) might lead to problematic score functions when the input is not sampled from the ground truth trajectories, while $x_0$-prediction (Ramesh et al., 2022) or velocity prediction (Salimans & Ho, 2022). Here, the experiments are all conducted with the data augmentation proposed in Sec. 3.2. $\varepsilon$-prediction refers to predicting the corrected noise $\varepsilon'$, as defined in Eqn. B. Directly fitting Gaussian noise $\varepsilon$ does not produce reasonable results and is not included in the table. The results indicate that $x_0$-prediction yields better performance. Furthermore, the results from InstructPix2Pix demonstrate that switching to $x_0$-prediction alone does not enhance performance, indicating that the improvements in InstructPix2Mask are from data augmentation rather than merely switching to $x_0$-prediction.

**Timestamp-dependent Data Augmentation.** In Table D, we investigate the impact of varying the intensity of data augmentation across timesteps. We compare the linearly increasing intensity with the constant intensity. The results indicate that the dynamic strategy outperforms the static one. This improvement is likely due to the fact that timestamp-dependent augmentation aligns more closely with the original diffusion formulation by introducing less noise as $t$ approaches 0, since our goal is to simulate the distribution shift.

| | $\varepsilon$-prediction | $x_0$-prediction |
|---|---|---|
| InstructPix2Pix | 56.42 | 53.39 |
| + ADDP | 59.64 | **66.19** |

Table C: Analysis on Training objectives.

| Intensity | oIoU |
|---|---|
| constant | 66.07 |
| linear | **66.19** |

Table D: Augmentation Scales.

**Analysis of Individual Augmentations** We compare the three types of augmentation, color, shape, and location corruptions, to the ground truth masks proposed in Sec. 3.2. As shown in Table E, we discovered that the augmentations have different effectiveness in enhancing diffusion-based perception models.

| | Color | Shape | Location | oIoU |
|---|---|---|---|---|
| ADDP (w/o Aug) | | | | 64.0 |
| Color Only | ✓ | | | 64.2 |
| Shape Only | | ✓ | | 64.5 |
| Location Only | | | ✓ | 65.9 |
| ADDP (All Augs) | ✓ | ✓ | ✓ | 66.2 |

Table E: Comparison of augmentation types for RIS.

## B.4 Additional Analysis on Marigold-based Depth Estimation

In addition to the analysis in Table 1, we analyze the separate effects of *contribution-aware timestep sampling* (Sec. 3.1) and *diffusion-tailored data augmentation* (Sec. 3.2) for Marigold-based depth

estimation. As shown in Table F, our proposed techniques from ADDP both improve Marigold step by step. Notably, our data augmentation enhances Marigold even though its perception quality (Fig. 3) does not show significant drops as RIS, indicating the vast existence of distribution shifts. This supports the effectiveness of our proposed techniques in addition to Sec. 4.3: both our *contribution-aware timestep sampling* (Sec. 3.1) and *diffusion-tailored data augmentation* (Sec. 3.2) are beneficial for diffusion-based perception.

| Method | ETH3D AbsRel↓ | ETH3D $\delta_1\uparrow$ | ScanNet AbsRel↓ | ScanNet $\delta_1\uparrow$ | NYUv2 AbsRel↓ | NYUv2 $\delta_1\uparrow$ | Diode AbsRel↓ | Diode $\delta_1\uparrow$ | KITTI AbsRel↓ | KITTI $\delta_1\uparrow$ | Average AbsRel↓ | Average $\delta_1\uparrow$ | Rank |
|---|---|---|---|---|---|---|---|---|---|---|---|---|---|
| Marigold (Ke et al., 2024) | 7.1 | 95.1 | 6.9 | 94.5 | 6.0 | 95.9 | 31.0 | 77.2 | 10.5 | 90.4 | 12.3 | 90.6 | 2.9 |
| +Sampling (Ours) | **6.3** | **96.1** | 6.4 | 95.3 | 5.7 | 96.1 | 30.6 | 77.2 | 10.3 | 90.4 | 11.9 | 91.0 | 2.0 |
| +Sampling+Aug (Ours) | **6.3** | **96.1** | **6.3** | **95.6** | **5.6** | **96.3** | **29.6** | **77.5** | **10.0** | **90.6** | **11.6** | **91.2** | **1.1** |

Table F: Ablation analysis on the effects of *contribution-aware timestep sampling* ("Sampling") and *diffusion-tailored data augmentation* ("Aug") for Marigold. Both of them contribute positively to diffusion-based perception.

### B.5 COMPARISON WITH SAMPLING WEIGHTS FROM GENERATIVE STUDIES

Our ADDP leverages the $c_t$ estimated from the perception statistics to control the sampling of timesteps. Studies on diffusion models designed for generative tasks also notice the benefits of better loss scaling or probability scaling of timesteps. The comparison between our ADDP and their weights is in Table G. For a fair comparison, we scale the probability of timestep sampling for all the methods and disable our data augmentations.

| Method | oIoU |
|---|---|
| Uniform (DDPM (Ho et al., 2020)) | 56.42 |
| Estimated $c_t$ (Ours) | **64.00** |
| P2 (Choi et al., 2022) | 57.90 |
| Min-SNR (Hang et al., 2023) | 26.13 |
| MLT (Go et al., 2024) | 55.60 |
| SpeedD (Wang et al., 2024) | 57.89 |
| Soft Truncation (Kim et al., 2022a) | 57.49 |

Table G: Comparison with Sampling Weights from Generative Models' Studies.

Such an improvement primarily arises from better alignment between our weights estimated from the perception statistics with the perception tasks, whereas the others are mainly considered from a generative task perspective. This demonstrates the effectiveness and necessity of our investigation on diffusion-based perception instead of fully relying on the results from generative tasks as guidance.

### B.6 COMPARISON WITH AUGMENTATION STRATEGIES FROM GENERATIVE STUDIES

The studies of generative models also notice the distribution shift between training and denoising, termed as "exposure bias" (Ning et al., 2024). Therefore, we compare the augmentation strategies with our method under a diffusion-based perception setting. Since TADA (Park et al., 2023) primarily focuses on the intensity of data augmentation, we replace our linear intensity schedule with that from TADA while keeping the identical set of augmentations: color, shape, and location. As shown in Table H, our method outperforms these related approaches

| Method | oIoU |
|---|---|
| w/o Augmentation | 64.00 |
| w/ Augmentation (Ours) | **66.19** |
| TADA (Park et al., 2023) | 65.76 |
| Epsilon Scaling (Ning et al., 2024) | 64.13 |

Table H: Comparison with Augmentation Strategies from Generative Models' Studies.

from a generative task perspective, indicating the uniqueness of our diffusion-based perception investigation.

### B.7 INFLUENCE OF SAMPLING STEPS ON OUR OBSERVATIONS

When observing the behaviors of diffusion models for perception tasks (Fig. 1), we primarily follow the setting of InstructPix2Pix (Brooks et al., 2023) and adopt 100 sampling steps. However, the number of sampling steps might influence the properties of a diffusion model. We analyze the number of sampling steps to check our observation. Specifically, we let an InstructPix2Pix baseline model trained on RefCOCO generate intermediate referring segmentation results at different sampling steps: 50, 100, and 200. The evaluation of IoU regarding timesteps is in Fig. E.

As shown in Fig. E, our observations are consistent across different sampling steps: **(1)** the contributions across timesteps are significantly uneven, with earlier steps contributing more than the later ones; **(2)** the distribution shift between training and denoising causes the performance drop at later

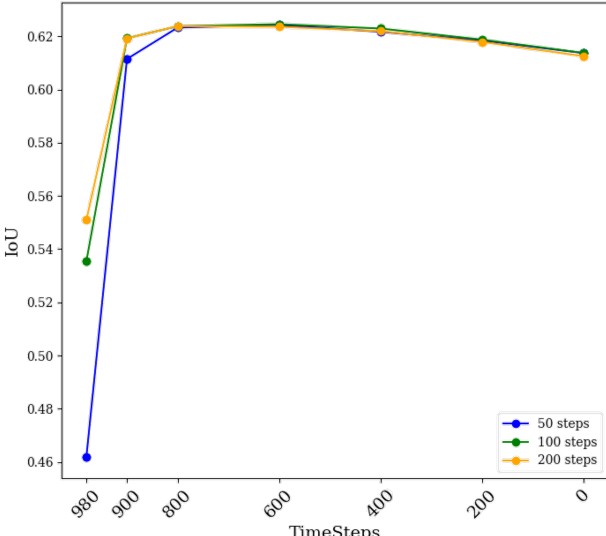

Figure E: Observations with different sampling steps.

denoising steps. When using fewer sampling steps, only the earlier steps suffer from insufficient diffusion denoising, but the performance quickly catches up after $t = 900$. These combined verified that our observed issues of diffusion-based perception are consistent.

