# OpenReview forum: "Aligning Generative Denoising with Discriminative Objectives Unleashes Diffusion for Visual Perception"
_ICLR.cc/2025/Conference — ICLR 2025 Poster_

### Official Review · Reviewer_ZiYy · 2024-10-27

**Soundness:** 3
**Presentation:** 3
**Contribution:** 2
**Rating:** 6
**Confidence:** 4

**Summary:**

This paper aims to improve the performance of repurposing the diffusion models as the deterministic task models. First, the authors analyze the performance changes through the denoising process of diffusion models in an RIS task, and they find that later timesteps are more influential to the mIOU but there is a decreasing tendency in mIOU when approaching the smaller timesteps. They hypothesize the reason for this finding as 1) the unequalized contribution of denoising steps and 2) the training-denoising distribution shift. To deal with the first challenge, they propose loss scaling and timestep sampling methods, which reflect varied contributions of each timestep for deterministic tasks. For the second challenge, data augmentation strategies suitable for each task are utilized to reduce exposure biases. Finally, they argue that the generative process in diffusion models is beneficial in reflecting users' feedback on perception tasks and propose agentic workflows that conduct multi-round interactions. The various experiments are conducted to show the effectiveness of their method mainly focusing on improving Marigold, InstructPix2Pix, and InstructCV. Technical details and additional experimental results are mainly presented in the Appendix.

**Strengths:**

- This paper deals with deep explorations of the denoising procedure of diffusion models in visual perception tasks.
- The main strength of this paper is the performance improvement. As shown in the results, the proposed methods offer substantial performance enhancement for various models.

**Weaknesses:**

- **W1-1) (Minor)** This paper introduces numerous techniques, but some details are obscured or inaccurately described. For example, in Eq. (2), $c_t^2$ is presented as a weight for each loss term based on the timestep, yet it is also used in the timestep sampling distribution in Fig. 2.
- **W1-2) (Major)** Furthermore, I cannot find the details of the analysis in Fig. 1. It is unclear how the authors calculated mIOU scores during the sampling process. If these scores are derived using only the estimated $x_0$​ values partway through the sampling, they don’t reflect the final generated results. Using these incomplete samples to support the authors’ conclusions seems insufficient. In other words, simply examining the denoising behavior of partial generations does not fully represent the actual generation process. A more accurate approach would be to inspect denoising characteristics across the complete generation process, as this would better validate the claims.
- **W2) (Major)** The technical contribution of this paper in the proposal of 1) a loss scaling and timestep sampling methods and 2) the dataset augmentation for each task can be trivial without comparing baselines.
  - **W2-1)** The authors introduce loss scaling and timestep sampling methods based on their observations of uneven contributions to performance. In the context of loss-weighting and timestep-sampling strategies within diffusion models [1, 2, 3, 4, 5], several studies have shown that emphasizing the loss at later timesteps can enhance model performance. The core operation of the authors’ method appears similar to these existing techniques. Therefore, without direct comparisons to these established methods, it is difficult to be convinced of any distinct advantage offered by the proposed approach in deterministic task setups. I suggest adding the comparative results with these methods.
  - **W2-2)** Similar to W2-1), [6] also deals with timestep-dependent data augmentation and it is necessary to compare this to validate the advantages of the proposed methods. Also, incorporating the data augmentation strategy for reducing exposure biases in diffusion models is already mentioned in [7]. In this regard, I suggest adding the comparative results with data augmentation works in diffusion model contexts.

- **W3) (Minor)** The proposed methods require a manual setup for data augmentation and $c_t$.

### **Reference**

[1] Perception Prioritized Training of Diffusion Models, CVPR 2022

[2] Efficient Diffusion Training via Min-SNR weighting Strategy, ICCV 2023.

[3] Addressing Negative Transfer in Diffusion Models, Neurips 2023

[4] A Closer Look at Timesteps is Worthy of Triple Speed-Up for Diffusion Model Training, Arxiv 2024.

[5] Soft Truncation: A Universal Training Technique of Score-based Diffusion Model for High Precision Score Estimation, ICML 2022

[6] TADA: Timestep-Aware Data Augmentation For Diffusion Models, Neurips 2023 workshop on Diffusion Models.

[7] ELUCIDATING THE EXPOSURE BIAS IN DIFFUSION MODELS, ICLR 2024.

**Questions:**

Q1: Why is linear regression needed in estimating $c_t$​ instead of determining $c_t$ directly?

---

> ### Author Response · Authors · 2024-11-24
> **Response to Reviewer ZiYy (Part 1/4)**
>
> We appreciate the reviewer's recognition of our deep exploration of diffusion-based perception and substantial improvement from our proposed framework. We additionally thank the reviewer for pointing to some related works from a generative task perspective.
>
> In the response below, we hope to answer the reviewer’s questions and clarify the distinctions of ADDP from a diffusion-based perception aspect. Because discriminative tasks require precisely fitting a rigorous ground truth, which is fundamentally different from generative tasks, our approach accordingly differs from the generative modeling ones. Still, we will gladly incorporate the related works and comparisons the reviewer suggested since they strengthen our discussion.
> For better clarity, we provide the response following the logical orders of the reviewer’s questions.
>
> ## 1. Analysis of Denoising Characteristics (W1-2)
>
> > It is unclear how the authors calculated mIOU scores during the sampling process.... A more accurate approach would be to inspect denoising characteristics across the complete generation process, as this would better validate the claims.
>
> The reviewer asks how we calculate the IoU scores for intermediate denoising steps in Fig. 1, and suggests “inspect denoising characteristics across the complete generation process.” In fact, our procedure is identical to the approach suggested by the reviewer: we first perform a full diffusion denoising process from $x_T$​ to $x_0$​, and then **extract the intermediate results from the same complete denoising trajectory**. Specifically, all the figures in Fig. 1(b) are generated from the same initial random noise $x_T$​ and **a single, complete denoising process**. The same process also applies to calculating the IoU of intermediate steps. We sincerely thank the reviewer for mentioning this and would gladly include this critical detail in the caption of Fig. 1.
>
> ## 2. $c_t^2$ Are Loss Weights or Sampling Weights? (W1-1)
>
> > In Eq. (2), $c_t^2$ is presented as a weight for each loss term based on the timestep, yet it is also used in the timestep sampling distribution in Fig. 2.
>
> In our Eqn. (2), the loss term is expressed as:
>
> $$
> \mathbb{E}_{(\tilde{x}_0, x_0, \epsilon_1, .., \epsilon_T)} [ \sum _{t=1}^T c_t^2 ||\epsilon_t - \epsilon _{ \theta }(x_t, t)||_2^2]
> $$
>
> Because it is an expectation $\mathbb{E}$ over many samples, $c_t^2$ being **either loss weights or sampling weights are equivalent**.
>
> As mentioned in Sec. 3.1 (L196-L200), we have experimented with using $c_t^2$ as both loss scaling or probability scaling. Our results in Table 5 show that both approaches lead to improvement, but **probability scaling performs better than loss scaling** (oIoU:  Probability scaling 63.05 v.s. Loss scaling 58.21). We conjecture that this is related to the stochastic gradient descent, where using more samples can lead to more stable gradients for the critical early denoising steps. Because of its better performance, our ADDP uses $c_t^2$ as sampling weights instead of loss weights for training.
>
> ## 3. Why Estimating $c_t$ Instead of Determining It Directly? (Q1)
>
> > Why is linear regression needed in estimating $c_t$ instead of determining $c_t$ directly?
>
> Following the discussion above, we answer the reviewer's question on why we **estimate** the values of $c_t$.
>
> First, **different perception tasks and metrics have divergent behaviors**, requiring different sets of $c_t$ to **reflect their individual properties**. For example, the depth estimation task shows a much smoother evolution of perception metrics during denoising than referring image segmentation (RIS), as in Fig. 3. This indicates that later steps should exhibit a larger contribution for depth estimation because depth estimation requires more detailed boundaries of objects for a small depth error. Therefore, estimating the $c_t^2$ from the statistics of perception metrics is necessary to capture the unique behaviors of perception tasks. Our estimated weights in Fig. A also demonstrate how $c_t$ significantly differs across different scenarios.
>
> Second, we agree with the reviewer that the weights of $c_t$ can be directly derived from the diffusion model formulations. In our paper, we call this variant “diffusion weights,” and it performs worse than our estimated weights** due to its worse alignment with the individual perception tasks. As experimented in Table 5, using the estimated $c_t^2$ outperforms the weights (oIoU: 64.00 vs 63.05). Therefore, the experiments also support using estimated weights for each perception task.
>
> To summarize, we propose to estimate $c_t^2$  **to reflect the uniqueness of diverse perception**, which is a major advantage and motivation of our method.

---

> ### Author Response · Authors · 2024-11-24
> **Response to Reviewer ZiYy (Part 2/4)**
>
> ## 4. Comparison with Loss Scaling Methods from Generative Tasks (W2-1)
>
> > The authors introduce loss scaling and timestep sampling methods based on their observations of uneven contributions to performance. ..  I suggest adding the comparative results with these methods.
>
> We start by comparing with the baselines suggested by the reviewer [1, 2, 3, 4, 5]. Concretely, we replace our estimated sampling weights $c_t$ with their approaches and have the performance in the following table.
>
> |        Methods         |   oIoU    |
> | :--------------------: | :-------: |
> |        Uniform Sample Baseline        |   56.42   |
> | Perception Stat (Ours) | **63.75** |
> |         P2 [1]         |   57.90   |
> |      Min-SNR [2]       |   26.13   |
> |        MLT [3]         |   55.60   |
> |       SpeeD [4]        |   57.89   |
> |  Soft Truncation [5]   |   57.49   |
>
> Based on the above results, we have the following observations and clarifications:
>
> * Our estimated $c_t$ shows a significant advantage.
>
> * Such an improvement primarily arises from better alignment between our weights estimated from the perception statistics with the perception tasks, whereas the suggested methods are mainly considered from a generative task perspective.
> * We hope such comparisons further demonstrate the effectiveness and necessity of our investigation on diffusion-based perception instead of fully relying on the results from generative tasks as guidance. We are grateful for the suggested papers from the reviewer and will incorporate them into our discussion in the current L169-L170.
>
> Furthermore, we believe that our ADDP also demonstrates fundamental differences compared with these generative studies.
>
> * The existence of rigorous ground truth in perception tasks supports us in **directly assessing the precision of every sample without considering diversity as in conventional generative tasks**. This property enables us to reveal that the contribution of timesteps varies across different tasks. In comparison, the suggested generative studies have not shown such distinctions.
>
> * We propose a method that automatically estimates the contribution factor $c_t$ based on the perception metric statistics, while the suggested methods pre-define the weights [4, 5], conduct hyper-parameter search *w.r.t.* FID [1, 2], or use loss as guidance [3].
>
> We hope these clarifications explain our uniqueness in estimating and utilizing the contribution facts compared with studies from a generative perspective. We thank the reviewer for suggesting related works on scaling the loss from a generative modeling perspective and will incorporate the above results in our revision for a thorough discussion.

---

> ### Author Response · Authors · 2024-11-24
> **Response to Reviewer ZiYy (Part 3/4)**
>
> ## 5. Comparison with Data Augmentations (W2-2)
>
> > Similar to W2-1), [6] also deals with timestep-dependent data augmentation and it is necessary to compare this to validate the advantages of the proposed methods. .. I suggest adding the comparative results with data augmentation works in diffusion model contexts.
>
> We appreciate the suggested paper from the reviewer and start our discussion on the effectiveness of data augmentation with the requested comparison.
>
> First, we compare our data augmentations with epsilon-scaling [7], as suggested by the reviewer. As shown in the table below, our augmentation exhibits a significant advantage over the method in [7]. We believe this marks the difference between discriminative perception tasks and conventional generative tasks, where perception tasks require stronger measures to address the training-denoising distribution shifts: **the shifted intermediate results might still produce reasonable images belonging to the ground truth distribution in a generative task, but they no longer align with the desired precise ground truth in a discriminative perception task**.
>
> |                     | oIoU |
> | ------------------- | ---- |
> | Baseline (w/ Timestep Sampling)            | 63.75 |
> | + Data Augmentation (Ours) | 66.19 |
> |  + Epsilon Scaling [7]  |64.13|
>
> Second, we compare it with the TADA paper mentioned by the reviewer. To the best of our knowledge, TADA has not yet released its code. So, we follow the curve in TADA’s Fig. 2 and the best hyper-parameters in TADA’s Table 8 and Table 9 for our timestep-dependent implementations. As shown below,  our method outperforms TADA under a discriminative perception scenario, marking the uniqueness of our diffusion-based perception investigation.
>
> |   Methods   |   oIoU    |
> | :---------: | :-------: |
> | ADDP (Ours) | **66.19** |
> |  TADA [6]   |   65.76   |
>
>
> Finally, we would like to clarify our main contribution and novelty in the data augmentation exploration: **By corrupting the ground truth, we can mitigate the training-denoising distribution shift for diffusion-based perception**. Therefore, the augmentations used in our work is also different from conventional perception and generative studies, as illustrated in the table below.
>
> | Tasks and Methods                          | Augmentations                                                |
> | ------------------------------------------ | ------------------------------------------------------------ |
> | RIS (Perception Augmentation)              | No Augmentations                                             |
> | RIS (Our Augmentation)                     | Change the *color*, *shape*, and *location* of ground truth mask |
> | Depth Estimation (Perception Augmentation) | *Horizontal flipping* of source image                        |
> | Depth Estimation (Our Augmentation)        | *Horizontal flipping* of the source image, *Gaussian blurring* for the ground truth depth map |
> | Generative Studies [7] | pixel blitting, geometric transformations, color transforms, image-space filtering, additive noise, and cutout for *the source image* |
>
> From this perspective, timestep-dependent augmentation is merely a design choice of our method. In fact, we experiment with both constant intensity and timestep-dependent augmentation in Table D, and a constant augmentation can outperform the baseline (oIoU 66.1 vs. 64.0) and is only slightly worse than the timestep-dependent method (oIoU 66.1 vs. 66.2).
>
> We hope the above discussion and comparison with the suggested papers from the reviewer clarify our uniqueness for diffusion-based perception and contributions.

---

> ### Author Response · Authors · 2024-11-24
> **Response to Reviewer ZiYy (Part 4/4)**
>
> ## 6. Timestep Sampling $c_t$ and Data Augmentation Needs Manual Setup (W-3)
>
> > The proposed methods require a manual setup for data augmentation and $c_t$.
>
> We appreciate the reviewer's concern about this. However, we believe that our methods require a similar amount of manual setup compared with most of the methods from the papers suggested by the reviewer.
>
> For the timestep sampling part, our $c_t$ is directly estimated from the statistics of a diffusion-based perception model. In comparison, the methods from the suggested papers require hyper-parameter searches: [1] and [2] propose sampling weights based on functions of SNR, but their approach requires manual tuning of hyper-parameters. [3] and [4] use statistics to determine the sampling weights, similar to our procedure.
>
> For the augmentation part, the only step that requires manual setup is searching the proper intensities of data augmentations, as shown in the main paper (Fig. 8). However, we suggest that such a hyper-parameter search is necessary for diffusion training. For example, the papers suggested by the reviewer also require hyper-parameter search to function, such as the $r_{rough}$ and $r_{fine}$ from [6] and the value of $b$ as in Table 15 and Table 16 from [7].
>
>
> [1] Perception Prioritized Training of Diffusion Models, CVPR 2022.
>
> [2] Efficient Diffusion Training via Min-SNR weighting Strategy, ICCV 2023.
>
> [3] Addressing Negative Transfer in Diffusion Models, NeuRIPS 2023.
>
> [4] A Closer Look at Timesteps is Worthy of Triple Speed-Up for Diffusion Model Training, Arxiv 2024.
>
> [5] Soft Truncation: A Universal Training Technique of Score-based Diffusion Model for High Precision Score Estimation, ICML 2022.
>
> [6] TADA: Timestep-Aware Data Augmentation For Diffusion Models, NeurIPS 2023 workshop on Diffusion Models.
>
> [7] Elucidating the Exposure Bias in Diffusion Models. ICLR 2024.

---

> > ### Comment · Area_Chair_BSt1 · 2024-11-24
> > **Discussion Period Ending Soon**
> >
> > Dear Reviewer,
> >
> > The discussion period will end soon. Please take a look at the author's comments and begin a discussion.
> >
> > Thanks, Your AC

---

> > ### Comment · Reviewer_ZiYy · 2024-11-25
> >
> > I sincerely appreciate the authors’ response. While parts of my concerns are resolved, the remaining concerns are not sufficiently addressed as follows:
> >
> > ### **1. Analysis of Denoising Characteristics.**
> >
> > Thank you for clarifying how the mIoU scores are calculated in Fig. 1. I understand now that the intermediate results and corresponding IoU values are derived from a single, complete denoising trajectory starting from the same initial random noise $x_T$. However, I still have some questions about this explanation. Since the diffusion sampling path is governed by a Stochastic Differential Equation (SDE), there is inherent randomness in the process. This implies that the final generated samples could vary even when starting from the same $x_T$. Could the authors elaborate on how they account for or mitigate the variability introduced by the stochasticity of the SDE in calculating intermediate IoU scores? For instance, do the reported results represent an average over multiple runs, or is a specific sampling strategy applied to ensure consistency?
> > Additionally, if the sampling was performed using an ODE process instead of an SDE, the analysis might only apply to specific characteristics of diffusion models under deterministic settings. Could the authors clarify whether the results are dependent on the sampling method and, if so, discuss any limitations this might introduce to the broader applicability of the findings?
> >
> > —
> >
> > Then, all of my other concerns are resolved. I sincerely appreciate your thorough response.

---

> > > ### Author Response · Authors · 2024-11-26
> > >
> > > We are glad to know that our clarifications are helpful in addressing your questions related to comparing our methods and related sampling weights and augmentation methods. Here, we provide more clarifications for your follow-up on our analysis.
> > >
> > > > if the sampling was performed using an ODE process instead of an SDE?
> > >
> > > Our referring image segmentation (RIS) strictly follows InstructPix2Pix and uses the “Euler ancestral” sampler from Karras *et al.*. **So it is a stochastic sampler.**
> > >
> > > > Could the authors elaborate on how they account for or mitigate the variability introduced by the stochasticity of the SDE in calculating intermediate IoU scores?
> > >
> > > Our analysis does not introduce special modifications from the conventional diffusion denoising process. The IoU-score curve from Fig. 1 is relatively stable, and the fluctuations are less than 0.4% from our three runs over the validation samples. The underlying reason for such a  small variance is that:
> > >
> > > * **Output format and evaluation protocol:** The perception tasks have a relatively simplified output space compared with conventional generative models. In our case, the model only learned to draw red masks in RIS tasks, and our evaluation further quantized pixels into binary masks via a color threshold. These combined make the variability of models less noticeable under a perception scenario.
> > > * **Classifier-free guidance weights:** We found that a large classifier-free guidance weight is crucial to achieving optimal perception performance. In practice, we applied $w_{\mathrm{txt}}=7.5$ and $w_{\mathrm{img}}=1.5$ for the text and image conditions, respectively. These guided the diffusion-based perception model to reflect the text and image conditions more faithfully instead of relying on its generative abilities. Therefore, large classifier-free guidance weights also contribute to the stability of predictions.
> > > * **Importance of earlier steps:** Another reason is that diffusion-based perception emphasizes the first few steps. The later steps do not exhibit significant changes or variability when determining the perception results.
> > >
> > > To demonstrate our clarifications more clearly, please refer to this [image provided in anonymous links](https://imgur.com/zb4kPOz). We show the intermediate results sampled from three different random seeds. As in the figure, sampling from different random seeds might have small variability due to stochasticity (marked in yellow circles), but their influence is minor compared with the overall perception results.
> > >
> > > > Could the authors clarify whether the results are dependent on the sampling method and, if so, discuss any limitations this might introduce to the broader applicability of the findings?
> > >
> > > As mentioned above, **our method follows the standard sampling methods in diffusion models** and does not explicitly rely on any specially modified sampler.

---

> > > > ### Comment · Reviewer_ZiYy · 2024-11-27
> > > >
> > > > The manuscript claims that diffusion-based perception emphasizes the first few steps, with later steps not exhibiting significant changes or variability when determining perception results. However, [A] presents a slightly different observation, suggesting that perceptual qualities are determined at middle timesteps. This raises some doubts about the accuracy or generalizability of your analysis, and I would appreciate further clarification or justification regarding this claim.
> > > >
> > > > Additionally, while the authors utilize a standard sampler, it’s worth noting that various hyperparameters, including the number of sampling steps, can significantly influence the results. Although the authors provide some details, I believe a more thorough discussion or analysis of the hyperparameter selection and its impact on the outcomes would strengthen the manuscript.
> > > >
> > > > ---
> > > >
> > > > Anyway, I guess that varying the configuration in the sampling method may not change the paper's observation. However, for the confirmation purpose, I suggest including these experiments in the final version. Before seeing these result, I would like to maintain my original rating.

---

> > > > > ### Author Response · Authors · 2024-11-27
> > > > > **Response to Reviewer ZiYy's Follow-Up (Part 1/2)**
> > > > >
> > > > > ## 1. Clarifications Regarding Paper [A]
> > > > >
> > > > > > However, [A] presents a slightly different observation, suggesting that perceptual qualities are determined at middle timesteps.
> > > > >
> > > > > We appreciate the reviewer for bringing this paper [A] up and would like to suggest that:
> > > > >
> > > > > * The observations in our paper and [A] do not contradict each other.
> > > > > * According to our experiments, with which the reviewer agrees, our weights estimated from the diffusion tasks perform better than those proposed in [A], from a generative task perspective.
> > > > > * The differences in tasks and models also determine that the weighting scheme for our approach and that of [A] cannot be identical.
> > > > >
> > > > > We provide a more detailed explanation as follows.
> > > > >
> > > > > First, the reviewer mentioned that [A] discovered "perceptual qualities are determined at middle timesteps," **but the full claim of [A]** (from the last paragraph of Sec. 3.1) is that "diffusion models learn *coarse features* (e.g., global color structure) at steps of *small SNRs*, *perceptually rich contents at medium SNRs*, and *remove remaining noise at large SNRs*." We clarify below that **the observations in our paper and [A] are not contradictory**.
> > > > >
> > > > > * In a perception scenario represented by our referring image segmentation (RIS), where the model learns to place the mask at the locations of target objects, the "global structure" plays a more critical role in the IoU than perceptual quality. This is verified by our observations and does not contradict the discovery in paper [A].
> > > > > * The final weighting scheme from [A] (Eqn. 8) demonstrates a similar trend to our weighting scheme: the earlier denoising steps (coarse stage, in [A]'s term) are assigned the largest weights, while the later denoising steps (content and clean-up stages, in [A]'s term) have smaller weights. Therefore, we suggest our observation is not contradicting [A]'s discovery.
> > > > > * The reviewer might be confused by [A]'s Figure 3 (c), which seems like the middle steps have higher weights. We would like to clarify that this is because [A] discusses loss in terms of $\frac{\beta_t}{(1-\beta_t)(1-\alpha_t)}||\epsilon - \epsilon _{\theta}(x_t, t)||^2$ (Eqn. 5 in [A]), while our weights are directly applied to $||\epsilon - \epsilon _{\theta}(x_t, t)||^2$ following DDPM. The trend of [A]'s loss weighting to DDPM's objective $||\epsilon - \epsilon _{\theta}(x_t, t)||^2$ is $\frac{1}{(k + \mathbf{SNR}(t))^{\gamma}}$ (according to [A]'s Eqn. 8). For your convenience, we normalize their weights and draw the curve following the style of Figure 3 in [A], as shown in [this image in anonymous link](https://imgur.com/a/SDG0IPX). [A]'s weights monotonically decrease when $t$ approaches 0 and $\mathbf{SNR}$ is larger. Therefore, [A]'s loss weighting scheme shows a similar trend as our weights.
> > > > >
> > > > > Second, we compared our weighting and [A]'s weighting in the previous response, according to the reviewer’s request. The results suggest that **our weighting scheme is better than [A]'s weighting scheme in quantitative comparison** (oIoU: 63.8 v.s. 57.9), because our weights are directly estimated from the perception statistics and can reflect the properties of perception tasks.
> > > > >
> > > > > Third, our weights are different from [A] because our settings and objectives are distinct:
> > > > >
> > > > > * Our perception scenarios always have a source image as input, which primarily regulates the variability of output, compared with the generative investigations from [A].
> > > > > * The perception scenario also focuses less on the "perceptual quality" of the output since the output is simply a red mask for the convenience of post-processing. Such characteristics of perception tasks also uniquely emphasize the earlier denoising steps.
> > > > >
> > > > > [A] Perception Prioritized Training of Diffusion Models, CVPR 2022.

---

> ### Author Response · Authors · 2024-11-27
> **Response to Reviewer ZiYy's Follow-Up (Part 2/2)**
>
> ## 2. Number of Sampling Steps
>
> > it’s worth noting that various hyperparameters, including the number of sampling steps, can significantly influence the results.... I guess that varying the configuration in the sampling method may not change the paper's observation. However, for confirmation purposes, I suggest including these experiments in the final version.
>
> Regarding the number of sampling steps, our RIS model follows InstructPix2Pix by using 100 inference steps out of the $T=1000$ total timesteps in a diffusion model. So, our implementation does not introduce any special tuning of the hyper-parameters.
>
> To fulfill the reviewer's request, we conduct a comparison between sampling with 50, 100, and 200 steps. Due to the limited time of the rebuttal, we randomly select 1000 samples from the approximately 10k samples of the RefCOCO validation set. In this [image from anonymous link](https://imgur.com/a/iHisWLk), we draw the IoU curve at intermediate denoising steps. As observed in the figure:
>
> * The conclusion in our paper is still consistent across all these timesteps: the early denoising steps ($t$ close to 1000) contribute significantly to the final results, and the degradation at later steps ($t$ close to 0) indicates the distribution shift.
> * When using fewer sampling steps, the very early denoising steps ($t=980$) have worse IoU results since they suffer from fewer denoising steps. However, the performance of different sampling steps is similar and approximately the same at $t=900$ and $t=800$, suggesting that our observations are stable.

---

> > ### Comment · Reviewer_ZiYy · 2024-11-27
> >
> > Apologies for the confusion in my previous response. The point I intended to raise is regarding your claim: "Another reason is that diffusion-based perception emphasizes the first few steps." This statement seems to somewhat contradict the observations in [A], and that was the main concern I was trying to convey. I now understand that you have already clarified this point, even though my earlier message may not have communicated my intent effectively.
> >
> > Additionally, I sincerely appreciate your thorough and thoughtful response. I would like to express my support for this paper and have updated my score accordingly. After synthesizing the feedback from all the reviewers, I feel that my concerns have been adequately addressed. Thank you again for your engagement and clarifications.

---

> > > ### Comment · Reviewer_ZiYy · 2024-11-27
> > >
> > > Also, I suggest including the experimental results and discussions in your revision.

---

> > > > ### Author Response · Authors · 2024-11-27
> > > >
> > > > Thank you again for your insightful review and detailed suggestions! We are glad to know that our clarifications and experiments have explained the uniqueness and effectiveness of our method for perception compared with generative studies. We have integrated your suggested discussions and experiments into our manuscript. We genuinely appreciate your discussion and your willingness to support our paper!

---

### Official Review · Reviewer_2CZ7 · 2024-10-31

**Soundness:** 2
**Presentation:** 3
**Contribution:** 3
**Rating:** 6
**Confidence:** 4

**Summary:**

This work proposes a method to bridge the gap between generative diffusion models and discriminative perception tasks by aligning the generative denoising process with perception objectives. One of its contributions is designing a loss function to reflect the varied contributions of timesteps. At the same time, it introduces diffusion-tailored data augmentation to simulate training-denoising distribution shifts and leverages the denoising process as an interactive interface for user correctional prompts. These enhancements improve performance without requiring architectural changes.

**Strengths:**

1. The paper presents a new way to make generative diffusion models work better for visual tasks by aligning them with discriminative tasks and gets SOTA performance.
2. It explains the differences between generative and discriminative processes and offers specific solutions like adjusting timestep sampling and using data augmentation designed for diffusion.
3. The method takes advantage of the interactive denoising process, allowing for correctional prompts and multiple rounds of interaction, which sets it apart from traditional models.

**Weaknesses:**

1. The data augmentation techniques used in this paper are quite standard for perception tasks and don't offer much innovation.
2. Although the paper demonstrates improvements in certain tasks, the applicability of these methods to a wider range of perception tasks and datasets still needs thorough validation.

**Questions:**

The proposed methods demonstrate strong performance on zero-shot benchmarks, but how robust and generalizable are these models in real-world applications? Are there additional experiments or analyses that validate the model’s stability across different environments and conditions?

---

> ### Author Response · Authors · 2024-11-24
> **Response to Reviewer 2CZ7 (Part 1/2)**
>
> We thank the reviewer for recognizing our improvement in diffusion for perception models via timestep sampling, data augmentation, and user interfaces proposed to align the denoising process and discriminative objectives. In addition, the reviewer provides some meaningful feedback and concerns, which we address below.
>
> ## 1. Contribution of Our Data Augmentation
>
> > 1. The data augmentation techniques used in this paper are quite standard for perception tasks and don't offer much innovation.
>
> Although we use the term "data augmentation," our data augmentation is very different from conventional perception studies, as we briefly discussed in Sec. 3.2 (L269 -- L274). Since the reviewer raises this concern, we would like to further clarify the uniqueness of our data augmentations from the perspective of diffusion-based perception.
>
> Our strategy significantly differs from previous perception studies in that **our augmentation corrupts the ground truth instead of the source images**. Such a design follows the principle of **simulating the distribution shifts caused by iterative sampling**, while conventional perception augmentations are mainly for the distribution shifts caused by training/inference data.
>
> Therefore, the data augmentation we employ is distinct from conventional perception methods. Concretely, *discriminative* referring image segmentation (RIS) methods actually cannot have any data augmentations because masks close to borders disable using "random cropping," and referring of "up/down/left/right" disables using "random flipping." We summarize the comparison of data augmentations for RIS and depth estimation below to show the difference. Hopefully, this table clarifies why our data augmentation differs from what is typical for perception tasks.
>
> | Tasks and Methods                          | Augmentations                                                |
> | ------------------------------------------ | ------------------------------------------------------------ |
> | RIS (Perception Augmentation)              | No Augmentations                                             |
> | RIS (Our Augmentation)                     | Change the *color*, *shape*, and *location* of ground truth mask |
> | Depth Estimation (Perception Augmentation) | *Horizontal flipping* of source image                        |
> | Depth Estimation (Our Augmentation)        | *Horizontal flipping* of the source image, *Gaussian blurring* for the ground truth depth map |
>
> In summary, our data augmentations are new explorations compared with perception-based augmentations in terms of both motivations and specific techniques.

---

> ### Author Response · Authors · 2024-11-24
> **Response to Reviewer 2CZ7 (Part 2/2)**
>
> ## 2. "Zero-shot Benchmarks" and "Real-world Tasks"
>
> > The proposed methods demonstrate strong performance on zero-shot benchmarks, but how robust and generalizable … across different environments and conditions?
>
> > Although the paper demonstrates improvements in certain tasks, the applicability of these methods to a wider range of perception tasks and datasets still needs thorough validation.
>
> The reviewer has concerns about whether our method applies to a broader range of perception tasks, while the other reviewers (apx4, zLxB, ZiYy) find our experiments thorough and offer improvement for various methods. From the reviewer’s question on “zero-shot benchmark” and “real-world tasks,” we guess the reviewer’s confusion might arise from the caption in Table 1. Below, we provide detailed clarifications of our settings and address the reviewer's concerns.
>
> First, we clarify that **our experiments have followed the real-world perception settings and covered multiple settings**. The reviewer's confusion might come from the "zero-shot benchmark" in the caption of Table 1. The sentence "improves Marigold across these zero-shot benchmarks" is meant to clarify that we have followed the exact setting of the previous state-of-the-art Marigold [1]. In fact, Marigold achieves competitive performance via first training on synthetic datasets and then transferring zero-shot to real-world benchmarks like NYUv2, ETH3D, etc. Our experiments follow the same setting for a fair comparison. From the reviewer’s question, we understand that the “zero-shot benchmark” might confuse readers unfamiliar with Marigold, and we have rephrased this sentence in the caption of the revision.
>
> Second, our evaluation has covered **three representative real-world perception scenarios** following the practice of state-of-the-art perception models: (1) Depth estimation, (2) Referring image segmentation (RIS), (3) Generalist model covering depth estimation, image segmentation, and object detection. Among them, depth estimation is based on a state-of-the-art depth estimator, Marigold [1]; RIS follows the standard practice, *e.g.*, UNINEXT [2], on RefCOCO, RefCOCO+, and G-Ref; and our generalist model follows InstructCV [3]. Therefore, our experiments have shown a comprehensive coverage of image-based geometric understanding, vision-language understanding, and generalist models, beyond prior work on this topic.
>
> Third, our experiments follow the standard implementations of real-world perception models, instead of zero-shot generalizing an image-editing InstructPix2Pix to various perception tasks. For example, our experiments on RIS fine-tuned the diffusion model on the training sets of RefCOCO, RefCOCO+, and G-Ref according to standard perception models.
>
> We hope the above explanation addresses the reviewer's concerns over the validity of our evaluation and improvement to a wide range of perception tasks. Please let us know if you have further concerns.
>
> [1] Ke et al. Repurposing Diffusion-Based Image Generators for Monocular Depth Estimation. CVPR 2024.
>
> [2] Yan et al. Universal instance perception as object discovery and retrieval. CVPR 2023.
>
> [3] Ga et al. InstructCV: Instruction-tuned text-to-image diffusion models as vision generalists. ICLR 2024.

---

> > ### Comment · Area_Chair_BSt1 · 2024-11-24
> > **Discussion Period Ending Soon**
> >
> > Dear Reviewer,
> >
> > The discussion period will end soon. Please take a look at the author's comments and begin a discussion.
> >
> > Thanks, Your AC

---

> ### Author Response · Authors · 2024-11-28
> **Follow-up for Reviewer 2CZ7**
>
> Dear Reviewer 2CZ7:
>
> Thank you for your time and effort in reviewing our manuscript! We have carefully addressed all of your comments in our previous response. If our revisions and clarifications have resolved your concerns, we kindly ask that you acknowledge this. If you have any additional questions or require further clarification, we would happily address them.
>
> Best,
>
> Authors of Submission 5156

---

> > ### Comment · Area_Chair_BSt1 · 2024-12-01
> > **Discussion Period**
> >
> > Dear Reviewer,
> >
> > Discussion is an important part of the review process. Please discuss the paper with the authors.
> >
> > Thanks, Your AC

---

### Official Review · Reviewer_zLxB · 2024-10-31

**Soundness:** 4
**Presentation:** 3
**Contribution:** 4
**Rating:** 8
**Confidence:** 4

**Summary:**

This paper investigates the performance gap between diffusion models and traditional encoder-decoder methods in perceptual tasks (e.g., Referring Segmentation) through validation experiments. It analyzes the varying contribution importance of different timesteps in perception through timestep-contribution analysis, leading to improvements in both training strategy and data aspects. Additionally, the paper demonstrates the potential of diffusion models for interactive referring perception tasks using prompt-based guidance agents.

**Strengths:**

1. The methodology and motivation are presented with exceptional clarity, with comprehensive consideration of variants for each method, including:
   - Various augmentation approaches
   - Different definitions of $c_{t}^2$ in timestep-aware training

2. The experimental section is thoroughly comprehensive, including:
   - Validation experiments in the introduction
   - Performance evaluations across multiple tasks (RIS, Depth Estimate, Generalist Perception)
   - Detailed ablation studies

3. The innovations stem from meaningful problem discovery and are non-trivial, with adaptive training and data augmentation for different timesteps representing logical explorations of the original problem.

**Weaknesses:**

1. The paper appears overly content-rich for a conference paper, resulting in some crucial discussions being relegated to supplementary materials:
   - The revised objective for the augmented ground truth in Sec. B.3.2
   - Intensity of augmentations across timesteps in Sec. B.3.1
   These aspects are fundamental to understanding the methodology but lack explanation in the main text.

2. The "User interface" section seems disconnected from the main methodological contributions and shows limited performance improvements (as shown in Table 3). The whole content of this paper might be better suited as two separate conference papers.

3. Concerns about the baseline comparison: It's unclear whether the InstructPix2Pix baseline in RIS underwent equivalent training. The paper doesn't specify if the baseline was trained on specific RIS datasets, raising questions about the fairness of comparing a general image editing model with a task-specific fine-tuned model.

**Questions:**

Please refer to the weaknesses section, particularly regarding:
1. The justification for baseline comparison methodology
2. The rationale behind including the user interface section in this paper

---

> ### Author Response · Authors · 2024-11-24
> **Response to Reviewer zLxB**
>
> We appreciate the reviewer recognizing our contribution and efforts to advance the alignment between diffusion models and perception tasks. We have enhanced diffusion models from three aspects: two focusing on training (**learning objective** and **training data** ) and one on inference (**user interface**). We hope the answers below can address the reviewer’s concerns.
>
> ## 1. Crucial Discussions in the Supplementary Materials
>
> > The paper appears overly content-rich for a conference paper, resulting in some crucial discussions being relegated to supplementary materials.
>
> We appreciate your suggestions and for pointing out that the paper is "content-rich." We indeed agree that our analysis of data augmentation details from Sec. B.3 could be meaningful for understanding the effectiveness of our design. According to your suggestions, we will revise Sec. 3.2 by explicitly mentioning the supplementary material’s conclusions and then pointing to the supplementary material. Please let us know if you have further suggestions, and we would be happy to incorporate them.
>
> ## 2. "User Interface" Section Seems Disconnected
>
> > The "User interface" section seems disconnected from the main methodological contributions and shows limited performance improvements (as shown in Table 3). The whole content of this paper might be better suited as two separate conference papers.
>
> First, we would like to clarify how our “user interface” is an integral component of the objective of this paper, which aims to improve and reveal the alignment between the diffusion denoising process and discriminative perception tasks.
>
> * The “user interface” section is critical because it addresses a fundamental yet unanswered question: *why are generative diffusion-based models aligned with perception tasks, given that we already have single-round and simpler discriminative models?* Our “user interface” proposal reveals that such a generative process is an interactive user interface – a capability not possessed by discriminative models. It demonstrates a direct alignment between the inference-time diffusion process and perception tasks.
>
> * Furthermore, our “user interface” perspective can seamlessly integrate with the previous two training-time improvements. It functions as the inference-time alignment between a diffusion process and perception tasks. On top of the models trained with our “contribution-aware timestep sampling” and “diffusion-tailored data augmentation,” this user interface has improved vision-language reasoning via correctional prompts automatically generated from foundation models, as shown in Table 3.
>
> Second, we would like to clarify more for Table 3, especially the significance of improvement of referring image segmentation (RIS).
>
> * Although it shows a smaller enhancement compared with our previous two training-time enhancements, the 1% IoU (from 55.85% to 56.98%) improvement on the most challenging RIS set, “G-Ref,” is already significant. As a reference, the 0.7 IoU increase from VPD [1] is already considered “consistent improvement.”
>
> * In addition, note that we use off-the-shelf vision-language models (GPT4o and LLaVA) to provide the correctional prompts so that we can evaluate them on a large scale. As existing off-the-shelf vision-language models frequently cannot correctly "guess" the confusion object, the bottleneck to our correctional guidance is the vision-language models. Therefore, we believe it will be more effective with guidance from real-human users and more advanced vision-language models in the future.
>
>
> ## 3. Clarifications on a Fair Comparison
>
> > It's unclear whether the InstructPix2Pix baseline in RIS underwent equivalent training.
>
> Regarding the question of "whether the baseline InstructPix2Pix is trained on the same RIS datasets," we clarify that **InstructPix2Pix is also finetuned on the RIS dataset** under the identical 60 epoch setting as our method for the results in Table 2. Thanks to your question, we will revise Table 2 to highlight our fair comparison: **InstructPix2Pix is finetuned for RIS** instead of using its original image editing weights.
>
> [1] Zhao et al. Unleashing Text-to-Image Diffusion Models for Visual Perception. ICCV 2023.

---

> > ### Comment · Reviewer_zLxB · 2024-11-25
> >
> > I appreciate the response from the authors. The rebuttal addressed most of my issues, including the over-riched content and the fair comparison. I would like to maintain my decision as a clear accept.

---

> > > ### Author Response · Authors · 2024-11-26
> > >
> > > Thank you again for your insightful review of our paper and for recognizing our contribution to diffusion-based perception! We are glad that our clarifications addressed your concerns on the critical role of diffusion models as perception “user interface” and the fairness of our comparisons.

---

### Official Review · Reviewer_apx4 · 2024-11-04

**Soundness:** 3
**Presentation:** 3
**Contribution:** 3
**Rating:** 6
**Confidence:** 4

**Summary:**

The paper "Aligning Generative Denoising with Discriminative Objectives Unleashes Diffusion for Visual Perception" introduces a novel framework, ADDP, designed to enhance the performance of generative diffusion models on discriminative visual perception tasks. These tasks, such as depth estimation, referring image segmentation, and generalist perception, demand precise alignment with ground truth data, which has been a challenge for generative models.

This technique prioritizes earlier denoising steps, which have a more significant impact on final perception quality. By adjusting the learning objectives, ADDP ensures that the model focuses on the most critical stages of the denoising process. To mitigate the "training-denoising distribution shift" issue, ADDP employs task-specific data augmentation strategies, simulating the deviations that occur during the denoising process to improve the model's robustness and maintaining perception quality across different timesteps.
ADDP leverages the iterative nature of diffusion models to enable user interaction. By incorporating classifier-free guidance, users can provide corrective feedback to refine the model's output, offering a more flexible and interactive approach compared to traditional discriminative models.

ADDP has demonstrated significant improvements in depth estimation, referring image segmentation, and generalist perception tasks. By effectively aligning generative and discriminative objectives, ADDP bridges the gap between generative diffusion models and discriminative baselines, without requiring architectural changes or additional data. This work represents a significant step forward in the field of generative diffusion models, highlighting their potential as powerful tools for visual perception tasks.

**Strengths:**

The paper introduces a novel framework, ADDP, addressing a previously unexplored gap in applying generative diffusion models to perception tasks. The paper's key contributions include a Contribution-Aware Timestep Sampling to prioritize early denoising steps for improved perception accuracy, as well as a Diffusion-Tailored Data Augmentation technique that simulates distribution shifts during the denoising process. The authors provide a Interactive User Interface to enable human-in-the-loop correction through the denoising process.

The paper demonstrates high quality through evaluation on various tasks, including depth estimation, referring image segmentation, and generalist perception and provides significant improvements over standard diffusion-based models without architectural changes.The authors provide ablation studies to justify the design choices and the individual contributions of each technique.
The paper is well-written and easy to understand, with clear structure, clear explanations of technical concepts, aided by diagrams and clear communication of key ideas.

The paper has significant implications in expanded model versatility, using for both generative and perception tasks and in possibilities for human-in-the-loop correction and user-guided generation. Overall, the paper presents a highly original, well-supported, and clearly articulated framework, expanding the utility of diffusion models into discriminative and perception tasks.

**Weaknesses:**

Not enough explanation about how the  contribution factors are derived and adapted to each perception task. It would be helpful if the authors could discuss the potential challenges or limitations of this augmentation strategy, particularly for tasks where shifts may not be well-simulated by data corruption.

The interactive correctional guidance using classifier-free guidance is a compelling feature, but  the authors could elaborate on how correctional prompts are formulated and whether they require manual input.The paper demonstrates improved results with diffusion-tailored data augmentation, but it is unclear whether specific types of augmentation (e.g., color, shape changes) have a greater impact than others without clarify which augmentations are most effective.

While ADDP is demonstrated with specific models, it’s unclear how generalizable these methods are across different diffusion model architectures, such as conditional or latent diffusion models with varying noise schedules. Although ADDP is tailored for perception, it would be interesting to know if the authors have considered its potential for other tasks, such as text-to-image generation or other generative tasks with precision requirements.

**Questions:**

1. Could the authors provide additional insights into how the contribution factors $𝑐_𝑡^2$   for each timestep are estimated and why they differ for various perception tasks? Specifically, how does the estimation process differ between depth estimation and referring image segmentation (RIS)?
2. Can the authors clarify how well the proposed augmentation strategy generalizes to other types of distribution shifts in diffusion-based perception tasks?
3.  How effective is the system without human-generated prompts?

---

> ### Author Response · Authors · 2024-11-24
> **Response to Reviewer apx4 (Part 1/5)**
>
> We appreciate the reviewer finding our framework novel and mitigating a previously unexplored gap of diffusion-based perception models. We are also grateful that the reviewer considers our evaluation "high-quality" by covering various tasks and achieving significant improvement. We thank the reviewer for the suggestions and provide our detailed response below. For the reviewer's ease of understanding and convenience, the answers are organized in a logical order.
>
> ## 1. Questions on Contribution Factors $c_t$
>
> We use **a unified and automatic strategy** to estimate the contribution factors $c_t$ for each diffusion-based perception task, and the different values of $c_t$ for varied tasks naturally reflect the unique properties of each task. We provide more detailed clarifications below.
>
> ### 1.1 Why $c_t$ Differs Across Tasks
>
> > Could the authors provide additional insights into why $c_t^2$ differ for various perception tasks?
>
> The reviewer asks why the contribution factors $c_t$ differ across diverse tasks, e.g., referring image segmentation (RIS) and depth estimation. We would like to clarify both quantitative and qualitative aspects:
>
> *  **Different contribution factors for varied tasks come from observing the denoising behavior of diffusion models trained on RIS or depth estimation datasets.** As in Fig. 3, RIS showed a more severe evolution of "perception metrics -- denoising steps" than depth estimation. Therefore, the $c_t$ should also be different to reflect the distinct contributions of timesteps for these two tasks.
> * **The difference of $c_t$ can also be explained by the varied properties of each perception task.** For instance, RIS emphasizes the global understanding and localization of objects, so its metric oIoU is mainly influenced by whether the model can find the correct object in the initial steps, leading to a higher $c_t$ at the first few denoising steps. However, depth estimation quality is evaluated via a per-pixel error, where the accuracy of local details and boundaries has significant effects. So, the later denoising steps can refine the depth estimation results and contribute more to the depth estimation quality. Combining the two aspects, the later denoising steps of depth estimation have a larger $c_t$ than RIS, consistent with our quantitatively estimated $c_t$.
>
> To summarize, **we design our ADDP to reflect the distinct properties of each perception task by estimating $c_t$ according to their own statistics**. We have shown the different estimated $c_t$ in Fig. A (Appendix), where RIS has a more uneven distribution than depth estimation.

---

> ### Author Response · Authors · 2024-11-24
> **Response to Reviewer apx4 (Part 2/5)**
>
> ### 1.2 How $c_t$ is Estimated and Used for Different Tasks?
>
> > Could the authors provide additional insights into how the contribution factors $c_t^2$ for each timestep are estimated?
>
> Following the discussion above, we have established **a unified and automatic way** to estimate $c_t$ and apply it to the diverse tasks. We briefly explained it in Sec. 3.1 (L209-L244) and provided the pseudo-code in Sec. A.2.2 (Appendix). We summarize the most important motivations and designs here to address your concerns.
>
> * Step 1: The $c_t$ for a perception task is estimated from a diffusion-based perception model trained on the perception data in a regular generative model’s way, e.g., InstructPix2Pix or Marigold. When running its diffusion denoising on an example, we collect the intermediate denoising results on the sampling trajectory, denoted as $x_T, ..., x_1, x_0$. According to the diffusion formulation, we can convert these $x_t$ into a "clean" prediction $x_t^0$ via the formula of Eqn. 3 as below. (Please refer to Fig. 1 (b) for a visualization.)
>
> $$
> x_t^0 = \frac{1}{\sqrt{\bar{\alpha}_t}}x_t - \frac{\sqrt{1 - \bar{\alpha}_t}}{\sqrt{\bar{\alpha}_t}} \epsilon _{\theta}(x_t, t)
> $$
>
> * Step 2: Calculate the perception metric $Q(\cdot)$ for these intermediate steps and final prediction, such as IoU for the RIS task.  We denote these intermediate perception metrics as $Q_T, ..., Q_1, Q_0$. In this way, we draw the curve in Fig. 1 (a) and Fig. 3.
>
> * Step 3: To quantify the contribution of a step $t$ for the final prediction, we utilize the concept of $R_2$: **coefficient of determination** from statistics. It represents the proportion that the input variables can explain dependent variables. Using $R_2$, we can run linear regression on the perception metrics $Q_{T:0}$ by treating the final prediction $Q_0$ as the dependent variable and $Q_{T:1}$ as the input variables. The resulting $R^2$ represents how much a denoising step $t$ accounts for the final prediction, which we use as $c_t^2$.
>
> * Step 4: After estimating $c_t^2$, we use the $c_t^2$ to scale the probability or loss of a timestep during denoising training as Sec. 3.1 for different tasks.
>
> As explained above, the tasks' distinctions are mainly considered using their perception metrics $Q(\cdot)$. That said, **we have used the same automatic estimation and utilization process for all the perception tasks** studied in the submission.
>
> When applying the above procedure to RIS and depth estimation, only the diffusion model and metrics $Q(\cdot)$ are changed:
>
> * RIS: using an InstructPix2Pix finetuned on the RIS data to produce intermediate step results, and IoU as the metric.
> * Depth estimation: using Marigold [1] to produce intermediate step results, and RMSE as the metric.
> * All the other estimation operations are the same for RIS and depth estimation.
>
> ## 2. Questions on Diffusion-tailored Data Augmentation
>
> ### 2.1 How Well Augmentation Generalizes to Other Distribution Shifts
>
> > Can the authors clarify how well the proposed augmentation strategy generalizes to other types of distribution shifts in diffusion-based perception tasks?
>
> In this paper, our augmentation strategy is proposed to address **the training-inference distribution shift caused by the iterative generation process**: $x_t$ during training time comes from $x_0$, while $x_t$ during inference time comes from iterative sampling from random noise $x_T$.
>
> However, our data augmentation can also address **the distribution shift between training and inference data** for perception tasks. This is demonstrated in Sec. 4.3.2 and Fig. 7 (curves of “InstructPix2Pix + Sampling” and “InstructPix2Pix + Sampling + Aug”), where the IoU improves 4% at the **initial denoising step**, which is primarily related to the data distribution shift.

---

> ### Author Response · Authors · 2024-11-24
> **Response to Reviewer apx4 (Part 3/5)**
>
> ### 2.2 Challenges of Our Diffusion-tailored Data Augmentation
>
> > It would be helpful if the authors could discuss the potential challenges or limitations of this augmentation strategy, particularly for tasks where shifts may not be well-simulated by data corruption.
>
> The reviewer asks questions about the limitations of our data augmentation, especially for tasks where distribution shifts cannot be well-simulated by data corruption. The **ideal solution** to training-denoising distribution shift is to **perfectly simulate such shifts** via using the $x_t$ sampled from random noises for training instead of the $x_t$ derived from the ground truth $x_0$. However, such a sampling pipeline would make the training computationally infeasible. So, we propose data augmentation as an **efficient surrogate** since perfectly simulating the distribution shift is not feasible in practice. Therefore, our augmentations are not perfect, but they can reasonably simulate the patterns of distribution shift and enable the diffusion-for-perception models to be robust to wrong $x_t$ and actively correct it, instead of entirely relying on the results from the previous denoising steps.
>
> A critical challenge is to identify the optimal combination of augmentations that effectively improve the diffusion-based perception model. With additional experiments described in the next section, we discover that different types of augmentation can lead to varied performance of the diffusion-based perception model. This is also relevant to the question from the reviewer on the individual effects of our augmentations: color, shape, and location changes.
>
> ### 2.3 Analysis of Individual Augmentations
>
> > The paper demonstrates improved results with diffusion-tailored data augmentation, but it is unclear whether specific types of augmentation (e.g., color, shape changes) have a greater impact than others without clarify which augmentations are most effective.
>
> Following the discussion in the previous section, we compare the three types of augmentation: color, shape, and location corruptions to the ground truth masks (illustration in Fig. 4 (b)). We incorporate the three augmentations individually into our ADDP, and the results are as follows. As demonstrated, changing the location of ground truth masks has the most significant benefit for the final performance, indicating that some data corruptions are more effective. We have included this set of new comparisons in our manuscript.
>
> | Augmentation             | oIoU$\uparrow$ |
> | ------------------------ | ---- |
> | No Aug                   | 64.0 |
> | Color + Location + Shape | 66.2 |
> | Color Only               | 64.2 |
> | Shape Only               | 64.5 |
> | Location Only            | 65.9 |

---

> ### Author Response · Authors · 2024-11-24
> **Response to Reviewer apx4 (Part 4/5)**
>
> ## 3. Questions on Correctional Prompts
>
> > The interactive correctional guidance using classifier-free guidance is a compelling feature, but the authors could elaborate on how correctional prompts are formulated and whether they require manual input.
> >
> > How effective is the system without human-generated prompts?
>
> The reviewer asks if our system would be effective without human prompts as correctional prompts. Yes, **our system is effective without human prompts** and **we have built an automatic system** providing correctional prompts using vision-language models (LLaVA and GPT4o) in the paper for scalable evaluation on thousands of samples, for the results in Table 3.
>
> First, the correctional prompt is mainly used for vision-language understanding in our paper (RIS), and it can be formulated as a “language description that refers to wrong target objects” so that our diffusion-based perception model could avoid such wrong predictions. By formulation, it can come from either humans or vision-language models in an agentic workflow (explained in the next paragraph).
>
> Second, we construct an automatic workflow for providing correctional prompts without humans, so that we can evaluate scalably on the validation sets of RIS comprising thousands of examples. Since the main objective of our paper is to introduce diffusion-based perception models, our agentic workflow is a simple proof-of-concept. It is mainly described in Sec. 3.3 (L311 - L 317) and Sec. A.4.2 (Appendix), and we summarize the key steps here for your convenience. (1) We use LLaVA to caption the image to describe all the objects in detail. (2) Given a referring, GPT4o is called to analyze the caption and guess the top 3 confusing similar but wrong objects for RIS. (3) Each of the top 3 objects guessed by GPT4o becomes a correctional prompt and leads a segmentation mask. The final result comes from the pixel-level majority voting of the masks.
>
> Third, our system is effective without human prompts by utilizing the above simple framework. As shown in Table 3, our correctional guidance leads to a statistically significant 1% improvement, especially on the most challenging G-Ref set. Meanwhile, we observe that current off-the-shelf vision-language models still struggle with hallucination (e.g., for our LLaVA captioning step), which bottlenecks the system's performance. Therefore, we believe that our diffusion-based perception can be more beneficial with correctional prompts from more advanced vision-language models in the future.
>
> ## 4. Questions on Additional Architectures
>
> > How generalizable these methods are across different diffusion model architectures, such as conditional or latent diffusion models with varying noise schedules.
>
> In the paper, we mainly focused on Stable Diffusion 1.5 and Stable Diffusion 2.0, which are the major backbones in existing diffusion-for-perception studies [1, 2, 3]. We would like to clarify that diffusion-for-perception requires pre-trained diffusion models, especially those text-to-image pre-trained ones on multi-modal data, as prior knowledge. Therefore, **diffusion-for-perception methods primarily follow the Stable Diffusion series and use latent diffusion models.**
>
> However, to address the reviewer's request on other models with varying schedulers, **we additionally experiment with the recent Stable Diffusion 3.0, a rectified-flow-based model**. We adopt the identical setting as our ablation studies in the paper. Both the "InstructPix2Pix + SD3.0" baseline and our ADDP are trained on the RefCOCO training set for 20 epochs and evaluated on the validation set of RefCOCO.
>
> As shown in the table below, our ADDP still significantly improves when using the SD 3.0 model, which is based on rectified flow. Due to the time and computation limit in the rebuttal phase, we trained SD3.0 for 20 epochs on the RefCOCO training set. At the same time, SD3.0 might require more training iterations and data to outperform the SD1.5 results in our paper due to its much larger model capacity.** However, the comparison already shows that our ADDP can effectively optimize a different diffusion model for perception objectives**.
>
> |                         | oIoU $\uparrow$ |
> | ----------------------- | --------------- |
> | InstructPix2Pix + SD3.0 | 34.44           |
> | + ADDP                  | 41.42           |

---

> ### Author Response · Authors · 2024-11-24
> **Response to Reviewer apx4 (Part 5/5)**
>
> ## 5. ADDP for Generative Tasks
>
> > Although ADDP is tailored for perception, it would be interesting to know if the authors have considered its potential for other tasks, such as text-to-image generation or other generative tasks with precision requirements.
>
> We appreciate the reviewer’s comment on broadening the impact of our work. As acknowledged by the reviewer, our approach was primarily designed to address perception tasks based on diffusion models, an emerging and underexplored problem. Compared to prior work in this area, we have conducted a more comprehensive investigation across various perception tasks, including depth estimation (geometric understanding), referring image segmentation (vision-language understanding), and a multi-task generalist model.
>
> We agree with the reviewer and recognize the potential of our approach for addressing other tasks. One key characteristic of such tasks is that they require rigorous ground truth, similar to perception tasks. As a result, our approach may not be directly applicable to typical image generation tasks. However, it could be relevant for generative tasks with relatively well-defined ground truth, such as super-resolution [4] and 3D reconstruction [5]. We hope our work will inspire further exploration in this direction, including its extension to additional tasks.
>
> [1] Ke et al. Repurposing Diffusion-Based Image Generators for Monocular Depth Estimation. CVPR 2024.
>
> [2] Ga et al. InstructCV: Instruction-tuned text-to-image diffusion models as vision generalists. ICLR 2024.
>
> [3] Geng et al. InstructDiffusion: A Generalist Modeling Interface for Vision Tasks. Arxiv 2023.
>
> [4] Saharia et al. Image Super-Resolution via Iterative Refinement. TPAMI 2023.
>
> [5] Hong et al. LRM: Large Reconstruction Model for Single Image to 3D. ICLR 2024.

---

> > ### Comment · Area_Chair_BSt1 · 2024-11-24
> > **Discussion Period Ending Soon**
> >
> > Dear Reviewer,
> >
> > The discussion period will end soon. Please take a look at the author's comments and begin a discussion.
> >
> > Thanks, Your AC

---

> ### Author Response · Authors · 2024-11-28
> **Follow-up for Reviewer apx4**
>
> Dear Reviewer apx4:
>
> Thank you for your time and effort in reviewing our manuscript! We have carefully addressed all of your comments in our previous response. If our revisions and clarifications have resolved your concerns, we kindly ask that you acknowledge this. If you have any additional questions or require further clarification, we would happily address them.
>
> Best,
>
> Authors of Submission 5156

---

> > ### Comment · Area_Chair_BSt1 · 2024-12-01
> > **Discussion Period**
> >
> > Dear Reviewer,
> >
> > Discussion is an important part of the review process. Please discuss the paper with the authors.
> >
> > Thanks, Your AC

---

> > ### Comment · Reviewer_apx4 · 2024-12-02
> >
> > I appreciate your comprehensive response to my feedback. The rebuttal has effectively addressed the main concerns I raised. With these matters resolved, I am happy to reaffirm my decision to accept the paper.

---

> > > ### Author Response · Authors · 2024-12-02
> > >
> > > Thank you again for your insightful discussion of our paper and recognizing our contribution to diffusion-based perception! We are glad that our clarifications and experiments have addressed your concerns about the contribution factors, diffusion-tailored data augmentation, correctional guidance, and the generalizability of our approach. We have integrated your suggestions and experiments into our manuscript. We genuinely appreciate your discussion and support for our paper!

---

### Author Response · Authors · 2024-11-24
**General Response to All the Reviewers**

We appreciate the feedback and suggestions received from all the reviewers. We are glad that the reviewers found our exploration of diffusion-based perception previously **unexplored** (apx4), **new** (2CZ7), **non-trivial** (zLxB), and **deep** (ZiYy). In addition, the reviewers recognize our **thorough evaluation** (apx4, zLxB, ZiYy), which validates the **substantial improvement** of our approach for diffusion-based perception models (apx4, zLxB, ZiYy). The reviewers also encourage our exploration as a logical expansion of the original problem (zLxB), with innovations **set itself apart from traditional methods** (2CZ7), and potentials for **implications in expanded model versatility** (apx4).

We address the individual concerns of the reviewers below. We appreciate the reviewers for pointing out relevant studies in the context of generation-oriented diffusion models or conventional discriminative models, such as timestep-sampling strategies and data augmentation. We will cite all of these works in our revised manuscript. In the individual responses below, we elaborate on the specific differences and provide corresponding experimental comparisons. These detailed discussions and empirical results demonstrate the fundamental distinctions and strengths of our approach for diffusion-based perception tasks. In light of this, here we would like to clarify the key challenges faced by diffusion-based perception models and our major contributions:

* **Learning Objective**: We notice the **uneven contribution of timesteps across the denoising process** and adjusted the timestep sampling weights during training. Unlike previous generative studies, we propose **an automatic approach of estimating such sampling weights from perception statistics** to reflect the unique properties of each perception task.

* **Training Data**: We mitigate the **training-denoising distribution shift** by **corrupting the ground truth perception data**to simulate the distribution shifts during training. Notably, **our data augmentations differ from existing discriminative and generative methods**.

* **User Interface**: We demonstrate the unique advantage of diffusion models to **interact with users or foundation models via correctional prompts**, compared with conventional discriminative models. Such a proposal answers the long-standing question of “how can the stochastic generative process be useful for discriminative tasks.”

We hope the clarifications above summarize the novel contributions of our study and the contexts of our discussion.

---

### Author Response · Authors · 2024-11-27
**Revised Manuscript**

We greatly appreciate the comments and discussions with the reviewers and have incorporated the new analysis and results from the discussion into the updated manuscript:

## For Reviewer apx4

* Additional experiments in Sec. B.3 (Appendix) on individual effectiveness of our augmentations.
* An additional sentence in Sec. 6 discussing generative tasks as future works.

## For Reviewer zLxB
* An additional sentence in Sec. 3.2 mentioning the key implementation details on our data augmentations that are discussed in the Appendix.
* Revising the caption of Table 2 on the fair comparison of our study.

## For Reviewer 2CZ7
* Revising the caption of Table 1, clarifying the setting of Marigold, and avoiding the confusion of “zeros-shot benchmarks.”

## For Reviewer ZiYy
* Revising the caption of Table 1 clarifying how we got the results of intermediate denoising steps.
* Adding Sec. B.5 and Sec. B.6 (Appendix) for the suggested comparisons with weighting schemes and augmentation methods from generative studies.
* Adding Sec. B.7 (Appendix) clarifying the consistency of our observations regarding the number of denoising steps.

Please let us know if you have any additional comments. We would be happy to discuss them and incorporate your feedback into the revision.

---

### Meta-Review · Area_Chair_BSt1 · 2024-12-21

**Metareview:**

The paper addresses the question of generative processes like diffusion actually perform on perception tasks where there is a singular ground truth. The paper demonstrates that unlike image generation, perception tasks often rely significantly more heavily on certain timesteps (e.g. for segmentation almost all the work is done in the first set of initial timesteps). Based on this finding, the authors propose to estimate the importance of each step for each perception task and reweight the training steps of the diffusion models based on these importances. The paper also finds that later steps can 'corrupt' or worsen the result due to the divergence of the prediction from the ground truth. The paper proposes a diffusion tailored augmentation to simulate this drift in training data. Additionally the authors propose that the diffusion process allows users to interact with the perception task in order to correct the process.

The paper is very well written and presents both interesting analysis which leads to interesting design improvements. The paper shows strong quantitative improvements for depth estimation, semantic segmentation, and object detection. While the evaluation could have been more thorough, I do not find significant weaknesses in the paper. Reviewers pointed out that the "per-perception task" loss is a potential weakness as well as potentially weak baselines.

I agree with the reviewers and advocate for acceptance.

**Additional Comments On Reviewer Discussion:**

All reviewers advocate for acceptance with two 6s and one 8. Reviewer ZiYy initially questioned the technical contribution as well as some of the details of the analysis; however in the rebuttal, the authors managed to address these concerns. Reviewer apx4 highlighted the potential that their method is specific to each perception task and therefore it's unclear how generalizable these techniques are across architectures or other tasks but was also convinced by the rebuttal.

---

### Decision · Program_Chairs · 2025-01-22

Accept (Poster)